# Learning Label Shift Correction for Test-Agnostic Long-Tailed Recognition

## Abstract

Long-tail learning primarily focuses on mitigating the label distribution shift between long-tailed training data and uniformly distributed test data. However, in real-world applications, we often encounter a more intricate challenge where the test label distribution is agnostic. To address this problem, we first theoretically establish the substantial potential for reducing generalization error if we can precisely estimate the test label distribution. Motivated by the theoretical insight, we introduce a simple yet effective solution called label shift correction (LSC). LSC estimates the test label distribution within the proposed framework of generalized black box shift estimation, and adjusts the predictions from a pre-trained model to align with the test distribution. Theoretical analyses confirm that accurate test label distribution estimation can effectively reduce the generalization error. Extensive experimental results demonstrate that our method significantly outperforms previous state-of-the-art approaches, especially when confronted with non-uniform test label distribution. Notably, the proposed method is general and complements existing long-tail learning approaches, consistently improving their performance.

## 1 Introduction

Long-tail learning has garnered significant attention due to its prevalence in real-world applications (Deng et al., 2021; Zhang et al., 2023). Its central challenge revolves around addressing the detrimental effects of label distribution shifts between training and test datasets. Most existing methods tackle this challenge through three key strategies: 1) data manipulation during training (Zhou et al., 2020; Kang et al., 2020) 2) enhancing representation learning (Zhong et al., 2021; Cui et al., 2021), and 3) optimizing unbiased loss functions (Menon et al., 2021; Ren et al., 2020). These methods aim to ensure that learned models excel not only in the majority class but also in handling the minority class.

In the realm of long-tail learning, a prevalent assumption is that test data adheres to a uniform label distribution (Cao et al., 2019; Cui et al., 2019; Jamal et al., 2020; He et al., 2022). However, real-world scenarios, such as those in autonomous driving and recommender systems, often challenge this assumption. Models trained on data from one area may be deployed in another area with a markedly different data distribution. Consequently, the label distribution of test data can diverge significantly from that of the training data (Alexandari et al., 2020; Garg et al., 2020; Zhang et al., 2022; Wei et al., 2023). This divergence may manifest in various forms, including a similar long-tailed distribution or an inverse distribution. Existing long-tail learning methods tend to falter in such situations.

To enhance model generalization, we delve into the task of test-agnostic long-tail learning in this paper. Recent efforts in this direction include LADE (Hong et al., 2021) which relies on the assumption of accessible true test label distribution—an impractical requirement in most real-world scenarios. SADE (Zhang et al., 2022), on the other hand, introduces three skill-diverse experts to simulate different label distributions. A weighted ensembling of these experts, guided by an unsupervised self-supervised learning objective on test data, is leveraged to tackle more possible distributions. In fact, the multi-expert ensemble has been demonstrated effective not only in test-agnostic but also in conventional long-tail learning (Wang et al., 2021b; Li et al., 2022). In this paper, we go beyond the confines of a multi-expert ensemble and introduce a versatile method that seamlessly integrates with existing long-tail learning techniques, consistently enhancing their performance.

To address the challenge of unknown test label distribution, we ground our approach in theoretical insights that explicitly link generalization error to the discrepancy between estimated and true label

distributions. We introduce a straightforward estimation method employing a shallow neural network within the framework of generalized black box shift estimation. Specifically, we train this estimator by simulating diverse label distributions using the training dataset. The neural network takes the predicted logits from any pre-trained model as input and learns to approximate the true label distribution of these constructed subsets of training data. During testing time, the neural network provides an estimation of the test label distribution, which we then use to adjust the pre-trained model's outputs accordingly. Empirically, we discover that the estimation accuracy can be compromised by inflated logits of tail classes. This inflation arises from the nature of class-balanced loss, which encourages overconfidence in tail classes. To mitigate this issue, we introduce a novel approach: clipping spurious model outputs. The clipping thresholds can be dynamically adjusted to adapt to varying test label distributions.

Our contributions are fourfold: **1)** We introduce a straightforward yet effective method, LSC, to address test-agnostic long-tail learning, capable of accurately estimating test label distributions. **2)** We establish the theoretical foundation to demonstrate the capability of our method to provide more precise test label distribution estimations and reduce generalization error. **3)** We confirm the efficacy of the proposed method on three benchmark datasets. **4)** Importantly, LSC is compatible with existing long-tail learning methods, consistently enhancing their performance in test-agnostic scenarios.

## 2 RELATED WORK

**Long-tail learning.** Many existing methods have been proposed to improve the performance when dealing with long-tailed training datasets. Broadly, these approaches address the long-tail problem through three primary avenues: 1) manipulating training data like re-sampling (Zhou et al., 2020), data augmentation (Ahn et al., 2023) or external data (Yang & Xu, 2020), 2) improving representation learning via two-stage training (Kang et al., 2020; Zhong et al., 2021) or self-supervised (contrastive) learning (Cui et al., 2021; Kang et al., 2021; Li et al., 2022; Zhu et al., 2022), and 3) optimizing unbiased loss functions (Cao et al., 2019; Menon et al., 2021; Ren et al., 2020; Samuel & Chechik, 2021). Ensemble learning can further boost performance and is compatible with these approaches (Xiang et al., 2020; Wang et al., 2021b). While each method shows strong empirical performance, they typically assume a uniform label distribution in the test dataset, which may not hold in real-world applications.

**Test-agnostic long-tail learning.** Test-agnostic long-tail learning has recently garnered substantial attention due to its practical relevance. Specifically, LADE (Hong et al., 2021) assumes the availability of test class distribution and uses it to post-adjust model predictions. SADE (Zhang et al., 2022) learns multiple skill-diverse experts by implicitly simulating various label distributions. During testing, it employs a weighted ensemble of these experts, optimizing their weights using a self-supervised contrastive learning objective. Recently, BalPoE (Aimar et al., 2023) extends the idea of using multiple skill-diverse experts and incorporates *mixup* (Zhang et al., 2018) to further enhance the performance. It is worth noting that the test-agnostic problem shares similarities with test-time training (TTT) (Wang et al., 2021a; Sun et al., 2020; Iwasawa & Matsuo, 2021) which aims to adapt the model using test data. While we consider classic TTT methods in this paper, including those used in SADE, our results demonstrate that such general methods do not improve long-tail learning performance. Unlike previous approaches that assume accessibility to the underlying test label distribution, our proposed method seamlessly integrates with existing long-tail learning techniques to address test-agnostic challenges.

## 3 LSC: LEARNING LABEL SHIFT CORRECTION

In this section, we first introduce the problem setups. We then present LSC, a simple yet effective method that learns to estimate label shifts from data coupled with theoretical performance guarantees.

### 3.1 PRELIMINARIES

**Problem Setting.** We address the general $K$-class classification problem, where $\mathcal{X}$ and $\mathcal{Y}$ represent input and output spaces, respectively. $\mathcal{D}_S$ and $\mathcal{D}_T$ denote the joint distribution $(\mathcal{X}, \mathcal{Y})$ for training and test data, respectively. In the context of test-agnostic long-tail learning, $\mathcal{D}_S$ follows a long-tailed label distribution. We possess the training dataset $S = \{(x_n, y_n)\}_{n=1}^N \sim \mathcal{D}_S^N$, where $N$ represents the total number of training data points across all $K$ classes. For clarity, we assume the classes are

sorted by cardinality in decreasing order, such that $n_1 \gg n_K$, with $n_k$ indicating the number of samples in the $k$-th class. The imbalance ratio of $S$ is expressed as $\frac{n_1}{n_K}$. In conventional long-tail learning, methods are designed to create models that excel on test data following a uniform label distribution. However, in practical scenarios, test data often deviates from this ideal. For instance, the actual test label distribution may mirror the training data, denoted as $\mathbb{P}_{\mathcal{D}_S}(Y = y) = \mathbb{P}_{\mathcal{D}_T}(Y = y), y \in [K]$. Alternatively, it could be inversely long-tailed, expressed as $\mathbb{P}_{\mathcal{D}_S}(Y = y) = \mathbb{P}_{\mathcal{D}_T}(Y = inv(y)), y \in [K]$, where $inv(\cdot)$ reverses the order of the long-tailed distribution. In this paper, we refrain from assuming any prior knowledge about the test label distribution. Our objective is to discover a *hypothesis* $h : \mathcal{X} \to [K]$ that can generalize effectively on test data with varying label distributions. To achieve this, we adopt a scoring function $f : \mathcal{X} \to \mathbb{R}^K$ and employ $f$ to derive a hypothesis $h$ through $h_f(x) = \arg\max_{y \in [K]} f_y(x)$ where $f_y(x)$ returns the predicted logit of sample $x$ for class $y$. We define $\widehat{\mathbb{P}}(Y = y \mid x) \propto \exp f_y(x)$. This allows us to view $\widehat{\mathbb{P}}(\cdot \mid x) = \left[\widehat{\mathbb{P}}(Y = 1 \mid x), \ldots, \widehat{\mathbb{P}}(Y = K \mid x)\right]$ as an estimate of $\mathbb{P}_{\mathcal{D}_S}(\cdot \mid x)$. From this perspective, $h_f(x) = \arg\max_{y \in [K]} \widehat{\mathbb{P}}(Y = y \mid x)$. Additionally, we denote $h^* := \arg\min_h \epsilon_T(h)$ as the Bayes decision function in test distribution and $\epsilon_T(h) := \mathbb{P}_{\mathcal{D}_T}(h(X) \neq Y)$. Consistent with prior works, we focus on the label shift problem and assume that $\mathbb{P}_{\mathcal{D}_S}(X \mid Y) = \mathbb{P}_{\mathcal{D}_T}(X \mid Y)$.

## 3.2 ADJUSTING MODEL PREDICTIONS HELPS REDUCE GENERALIZATION ERROR

Denote $\mathbb{P}_{\mathcal{D}_S}(Y = y)$ and $\mathbb{P}_{\mathcal{D}_T}(Y = y)$ the label distributions of train and test domain, respectively. We construct the "post-adjusted" model outputs for each sample $x$:

$$\widetilde{f}_y(x) = f_y(x) + \log\left(\frac{\mathbb{P}_{\mathcal{D}_T}(Y = y)}{\mathbb{P}_{\mathcal{D}_S}(Y = y)}\right), \quad y \in [K] \tag{1}$$

If we can accurately estimate $\mathbb{P}_{\mathcal{D}_T}(Y = y)$, we can seamlessly adapt pre-trained models to the specific test dataset using Eq. (1). Starting with an estimated test label distribution $\widehat{\mathbb{P}}_{\mathcal{D}_T}(Y)$, our adapted model $\widetilde{f}$ induces a hypothesis as follows:

$$h_{\widetilde{f}}(x) = \arg\max_{y \in [K]} f_y(x) + \log\left(\frac{\widehat{\mathbb{P}}_{\mathcal{D}_T}(Y = y)}{\mathbb{P}_{\mathcal{D}_S}(Y = y)}\right) = \arg\max_{y \in [K]} \widehat{\mathbb{P}}(y \mid x) \frac{\widehat{\mathbb{P}}_{\mathcal{D}_T}(Y = y)}{\mathbb{P}_{\mathcal{D}_S}(Y = y)} \tag{2}$$

We then attempt to understand the effectiveness of the above optimizations through theoretical analysis. Performance guarantees are provided to limit the error gap between $\epsilon_T(h_{\widetilde{f}})$ and $\epsilon_T(h_T^*)$. We first explain some terms as follows. Given a scoring function $f$ and estimated posterior probability $\widehat{\mathbb{P}}(y \mid x)$ induced by $f$, we define the expected $L^1$ distance between $\widehat{\mathbb{P}}(Y \mid X)$ and $\mathbb{P}_{\mathcal{D}_S}(Y \mid X)$ as $\left\|\widehat{\mathbb{P}}(Y \mid X) - \mathbb{P}_{\mathcal{D}_S}(Y \mid X)\right\|_{L^1} = \mathbb{E}_{x \sim \mathcal{D}_S}\left[\sum_{y \in [K]} \left|\widehat{\mathbb{P}}(y \mid x) - \mathbb{P}_{\mathcal{D}_S}(y \mid x)\right|\right]$. This measures how well the predicted posterior probability approximates the true posterior probability. Building upon this, we define the weighted generalization of this term: $\left\|\widehat{\mathbb{P}}(Y \mid X) - \mathbb{P}_{\mathcal{D}_S}(Y \mid X)\right\|_{L^1,w} = \mathbb{E}_{x \sim \mathcal{D}_S}\left[\sum_{y \in [K]} w_y \left|\widehat{\mathbb{P}}(y \mid x) - \mathbb{P}_{\mathcal{D}_S}(y \mid x)\right|\right]$. This weighted metric incorporates the ratios $w \in \mathbb{R}^K$ to measure the approximation quality. Additionally, we define the *balanced posterior error* as $\text{BPE}(h_f) = \max_{y \in [K]} \mathbb{P}_{\mathcal{D}_S}(h_f(X) \neq y \mid Y = y)$. This quantifies the maximum probability of misclassification for our adapted model. Finally, we denote the $L^1$ distance between $\widehat{\mathbb{P}}_{\mathcal{D}_T}(Y)$ and $\mathbb{P}_{\mathcal{D}_T}(Y)$ as $\left\|\widehat{\mathbb{P}}_{\mathcal{D}_T}(Y) - \mathbb{P}_{\mathcal{D}_T}(Y)\right\|_{L^1} = \sum_{y \in [K]} \left|\widehat{\mathbb{P}}_{\mathcal{D}_T}(Y = y) - \mathbb{P}_{\mathcal{D}_T}(Y = y)\right|$.

**Theorem 3.1** (Error gap between adjusted classifier and Bayes-optimal classifier). *Given an estimated label distribution of test data $\widehat{\mathbb{P}}_{\mathcal{D}_T}(Y)$, a pre-trained scoring function $f$, and a hypothesis $h_{\widetilde{f}}$ induced by $\widetilde{f}$, then the error gap $\epsilon_T(h_{\widetilde{f}}) - \epsilon_T(h^*)$ is upper bounded by:*

$$\left\|\widehat{\mathbb{P}}(Y \mid X) - \mathbb{P}_{\mathcal{D}_S}(Y \mid X)\right\|_{L^1,w} + \text{BSE}(h_f) \left\|\widehat{\mathbb{P}}_{\mathcal{D}_T}(Y) - \mathbb{P}_{\mathcal{D}_T}(Y)\right\|_{L^1}, \tag{3}$$

*where the ratio $w = \left(\frac{\widehat{\mathbb{P}}_{\mathcal{D}_T}(Y=1)}{\mathbb{P}_{\mathcal{D}_S}(Y=1)}, \frac{\widehat{\mathbb{P}}_{\mathcal{D}_T}(Y=2)}{\mathbb{P}_{\mathcal{D}_S}(Y=2)}, \cdots, \frac{\widehat{\mathbb{P}}_{\mathcal{D}_T}(Y=K)}{\mathbb{P}_{\mathcal{D}_S}(Y=K)}\right)$.*

**Remark.** Theorem 3.1 provides valuable insights by breaking down the upper bound of the error gap between the adjusted classifier and Bayes-optimal classifier into two terms. The first term characterizes the extent to which the posterior distribution predicted by the model approximates the true posterior distribution of the training data. Importantly, this term is solely related to the model's learning effectiveness on training data. It represents the inherent error associated with using $f$ to make predictions and is not influenced by label distribution shifts. The second term quantifies how closely the estimated test label distribution aligns with the true distribution. This term directly demonstrates that the upper bound of the generalization error on test data benefits from adjusting the model using a well-estimated label distribution. Furthermore, this term positively correlates with $\text{BSE}(h_f)$, which reflects the performance of $h_f$. The analysis also motivates us to train the scoring function $f$ by optimizing a class-balanced loss. We provide the proof in Appendix A.

**Generalized black box shift estimation (GBBSE).** Motivated by the insights from Theorem 3.1, our first task is to estimate the label distribution of the test domain. This can be formulated as the process of recovering the label distribution from an unlabeled testing dataset of size $M$: $T_M = \{(x_m, \cdot)\}_{m=1}^{M} \sim \mathcal{D}_T^M$.

In previous work, *black box shift estimation* (BBSE) (Lipton et al., 2018) has been a valuable technique for this purpose. We introduce $\widehat{M}_k, k \in [K]$ to represent the number of samples predicted as class $k$ by $h_f$, and construct a pseudo-label distribution $\widehat{\mathbb{P}}_{\mathcal{D}_T}(h_f(T_M)) = (\frac{\widehat{M}_1}{M}, \cdots, \frac{\widehat{M}_K}{M})^\top$. BBSE leverages this pseudo-label distribution and a conditional confusion matrix $\widehat{C}$ to estimate:

$$\widehat{\mathbb{P}}_{\mathcal{D}_T}(Y) = \widehat{C}_{h_f(X)|Y}^{-1} \widehat{\mathbb{P}}_{\mathcal{D}_T}(h_f(T_M)), \tag{4}$$

where $\widehat{C}$ is estimated from validation data and $\widehat{C}^{-1}$ denotes the inverse of $\widehat{C}$. Proposition 2 and Theorem 3 in Lipton et al. (2018) provide performance guarantees of BBSE (*i.e.,* the former ensures that BBSE estimators are consistent and the latter addresses the convergence rates).

In the context of long-tail learning, estimating the confusion matrix becomes challenging when dealing with limited observations for tail classes. To address this challenge, we draw inspiration from techniques like re-sampling and data augmentation, which are powerful tools for small sample parameter estimation. To use these tools in label distribution estimating, we introduce a *generalized black box shift estimation* (GBBSE). GBBSE takes a broader approach by introducing $\mathcal{D}_L$ as a joint distribution on the label distribution space $\Pi$, where $\Pi = \{\pi = (\pi_1, \pi_2, \cdots, \pi_K) \mid \sum_{k=1}^{K} \pi_k = 1, \pi_k \geq 0\}$). Similarly, we define a pseudo-label distribution space $\widetilde{\Pi}_M = \{\widetilde{\pi} = \widehat{\mathbb{P}}_{\mathcal{D}_T}(h_f(T_M)) \mid T_M \in \mathcal{X}^M\}$. GBBSE aims to find an estimator $g$ that minimizes the loss function $\epsilon_L(g) = \mathbb{E}_{(\pi, \widetilde{\pi}) \sim \mathcal{D}_L} [\ell(\pi, g(\widetilde{\pi}))]$, which $\ell$ measures the discrepancy between the estimated label distribution and the true label distribution. This is achieved through a family of parameterized label distribution estimation functions $g_\theta : \widetilde{\Pi}_M \to \Pi$, where $\theta$ is a parameter within a parameter space $\Theta$. GBBSE seeks to find the optimal parameter $\theta^* = \arg\min_{\theta \in \Theta} \epsilon_L(g_\theta)$ by training models on the training sets constructed through the re-sampling of multiple subsets from $\widetilde{S}$. Essentially, this involves augmenting the dataset $S$ with various label distributions. Indeed, the selection of $\ell$ can be arbitrary. In this paper, we employ the two most common metrics, *i.e.,* mean squared error and KL-divergence, and the impact of the selection of $\ell$ is studied in Table 13.

**BBSE as a special case of GBBSE.** It is known that on the training dataset $(X_S, Y_S)$, BBSE can achieve zero loss in recovering the pseudo-label distribution $\widehat{\mathbb{P}}(h_f(X_S))$ to label distribution $\widehat{\mathbb{P}}(Y_S)$. Now, consider a scenario where we use a single linear layer neural network with $K \times K$ parameters to implement the estimator and sample with $K - 1$ different class priors. In this situation, there exists a unique set of parameters that results in zero empirical recovery loss and GBBSE degenerates to BBSE. This observation implies that by generating a large $\widetilde{S}$, and directly optimizing $\epsilon_{\widetilde{S}}(g)$, we can further improve the generalization performance of $g_\theta$. Moreover, our experiments indicate that GBBSE can provide a more precise label distribution estimation compared to BBSE.

### 3.3 A SIMPLE INSTANTIATION OF GBBSE

We present LSC as a concrete instantiation of GBBSE, which comprises a *neural estimator* and *logit clipping*. LSC utilizes a neural network, specifically a multi-layer perceptron, to predict label

---

**Algorithm 1:** Meta algorithm to handle test label distribution shift

---

**input** Training data: $(X_S, Y_S)$, unlabeled test data: $X_T$, pre-trained model $f$

    ▷ `Sample from training data by varying class priors for` $Q$ `times`

1: Initialize $\widetilde{S} = \emptyset$

2: **for** $q = 1$ to $Q$ **do**

3:     $(\widetilde{X}, \widetilde{Y}) \leftarrow \text{SampleByClassPrior}(X_S, Y_S, \pi^q)$

4:     Compute class-wise average logits by $\widetilde{Z} = f(\widetilde{X})$ and $\widetilde{z}_{\text{avg}} = \frac{1}{|\widetilde{X}|} \sum_{i=1}^{|\widetilde{X}|} \widetilde{Z}_i$

5:     $\widetilde{S} = \widetilde{S} \cup (\widetilde{z}_{\text{avg}}, \pi^q)$

6: **end for**

7: Train *NeuralEstimator* $g_\theta$ on $\widetilde{S}$ by minimizing $\mathcal{L}(\widetilde{S}, g_\theta) = \frac{1}{|\widetilde{S}|} \sum_{(\widetilde{z}, \pi^q) \in \widetilde{S}} \ell(\pi^q, g_\theta(\widetilde{z}))$

8: Obtain predicted logits for test data using the pre-trained model by $Z_T \leftarrow f(X_T)$

9: Apply adaptive logits clipping on $Z_T$ with the value of $k$ set by Eq. (5) and obtain $\widehat{Z}_T$

10: Estimate test label distribution by $\widetilde{\pi} \leftarrow g_\theta(\widehat{z}_T)$, where $\widehat{z}_T$ is the class-wise average of $\widehat{Z}_T$

**output** Adjusted predictions $\widehat{Y}_T = \arg\max(Z_T + \log \widetilde{\pi})$

---

distribution, which is referred to as the *neural estimator*. We learn *neural estimator* on simulated subsets of training data following various class priors. For each predefined class prior $\pi^q$, we begin by sampling a subset of $M$ training data points. Specifically, we select $\lfloor M * \pi_k^q \rfloor$ training data points for class $k$. We then obtain predicted logits denoted as $Z$, from a pre-trained model $f$. The next step involves feeding the averaged class logits into *neural estimator*. The averaging is performed over the selected data points. The objective is to recover the true class prior $\pi^q$ using the averaged logits. By repeating this process with a range of predefined class priors, we can account for various possibilities of test label distributions. In implementation, we choose the imbalance ratio from the range of $[1/100, 100]$ and generate class priors for simulation.

In practice, $f$ can be realized by many long-tail learning methods such as RIDE (Wang et al., 2021b) and PaCo (Cui et al., 2021). Intriguingly, we discover that these methods tend to produce overconfident logits for tail classes while inhibiting head classes. The bias towards the tail classes can lead to undesirable label distribution predictions by *neural estimator*. To rectify the bias, we introduce *logit clipping*, which truncates small predicted logits for each sample to zero. The parameter $k$ controls how many of the smallest logits are clipped to zero. While we could treat $k$ as a tunable hyperparameter, we find that employing a fixed $k$ may not effectively adapt to varying test label distributions. To address this challenge, we propose an adaptive way of choosing $k$. Specifically, we determine $k$ based on a comparison between head and tail classes:

$$k = \arg\max_{k \in \mathcal{K}} \mathbb{I}(\pi_0^h > \lambda \pi_0^t)\widehat{Z}^h + \mathbb{I}(\pi_0^h < \lambda \pi_0^t)\widehat{Z}^t \quad s.t. \quad \widehat{Z} = logitClip(Z, k) \tag{5}$$

Here, $Z$ is the predicted logits by pre-trained model $f$ and $\mathcal{K}$ is the set of candidate values for $k$. $\pi_0$ denotes a direct estimate of label distribution derived from $\widehat{Z}$. We use $\pi_0^h$ and $\pi_0^t$ to represent the sum of head and tail classes in $\pi_0$, respectively. $\widehat{Z}^h$ and $\widehat{Z}^t$ are defined in a similar way to denote the sum of head and tail parts in $\widehat{Z}$, respectively. We use $\lambda > 1$ to control the level of ratio between $\pi_0^h$ and $\pi_0^t$. We set $\lambda = 1.5$ in all experiments and investigate the effect of different $\lambda$ values in the supplementary material. Nevertheless, we confirm the test label distribution as uniform if both conditions in Eq. (5) do not hold. The key steps of the method are summarized in Algorithm 1.

Additionally, we delve into the analysis of the Bayes error associated with recovering the pseudo-label distribution to the actual label distribution when implementing $\ell$ as the mean squared error. This error converges to zero at linear convergence rates, which ensures that we can use a parameterized function to estimate label distribution precisely when we have a large test set.

**Theorem 3.2** (Bayes error when using pseudo-label to estimate label distribution). *Given a hypothesis $h_f$, let $C_{h_f(X)|Y} \in \mathbb{R}^{K \times K}$ denote the conditional confusion matrix, i.e., $C_{h_f(X)|Y}(i, j) = \mathbb{P}(h_f(X) = i \mid Y = j)$. Suppose $C_{h_f(X)|Y}$ is invertible and the test label distribution $\pi$ is sampled uniformly at random from $\Pi$, then the error of Bayes function $g^*$ holds following inequality:*

$$\frac{K-1}{K(M+K+1)} \le \epsilon_L(g^*) \le \frac{K-1}{K(M+K+1)|\det(C_{h_f(X)|Y})|\sigma_{min}^2} \tag{6}$$

**Remark.** Theorem 3.2 ensures that we can acquire sufficient information about label distribution from the pseudo-label distribution, even when $h_f$ exhibits inherent errors. Suppose the error gap $\epsilon_L(g_\theta) - \epsilon_L(g^*)$ can be bounded through training $g_\theta$ using generated training set (it can be ensured by the Bayes-risk consistency of the training), then we can get a precise label distribution to adjust the model when the test sample size is large enough. Build upon this, the second term in the upper bound of $\epsilon_T(h_{\widetilde{f}}) - \epsilon_T(h^*)$ (as presented in Theorem 3.1) decreases. In addition, the term $\frac{1}{\sigma_{min}^2 |\det(C_{h_f(X)|Y})|}$ reflects the information loss when approximating ground-truth labels of test samples by pseudo-labels predicted from $h_f$. It is a direct way to decline this loss through reducing $\mathrm{BSE}(h_f)$ because we have $\sigma_{min} \geq 1 - 2\mathrm{BSE}(h_f)$. It also indicates that we should train the scoring function $f$ by optimizing the class-balanced loss instead of conventional cross-entropy. The proof is shown in Appendix B.

# 4 EXPERIMENT

## 4.1 EXPERIMENT SETUPS

**Datasets.** We conduct experiments using three widely used long-tail learning datasets, *i.e.,* CIFAR100-LT (Cao et al., 2019), ImageNet-LT (Liu et al., 2019), and Places-LT (Liu et al., 2019). On each dataset, we test the methods by simulating different test label distributions with varying imbalance ratios to assess the effectiveness of LSC in handling test-agnostic long-tail learning scenarios. The imbalance ratio (IR) is defined as $\mathrm{IR} = n_{max}/n_{min}$, where $n_{max}$ denotes the number of samples in the most frequent class, $n_{min}$ denotes the number of least frequent one. In particular, the **CIFAR100-LT** dataset is derived from the classic CIFAR100 dataset via subsampling, which has an imbalance ratio of 100, with $n_{max} = 500$ and $n_{min} = 5$. Similarly, the **ImageNet-LT** dataset is obtained by sampling from the popular ImageNet dataset with an imbalance ratio of 256. The **Places-LT** dataset is generated from the Places365 dataset, with an imbalance ratio of 996, and $n_{max} = 4980$, $n_{min} = 5$.

**Baseline.** We compare LSC with prior state-of-the-art long-tail learning approaches such as Balanced Softmax (Ren et al., 2020), PaCo (Cui et al., 2021), RIDE (Wang et al., 2021b), NCL (Li et al., 2022), SHIKE (Jin et al., 2023), and test-agnostic long-tail learning approaches including LADE (Hong et al., 2021), SADE (Zhang et al., 2022), BalPoE (Aimar et al., 2023).

**Evaluation protocols.** In the evaluation of test-agnostic long-tailed recognition, the models are tested on multiple test datasets, each with a different label distribution. These evaluations are performed following protocols established by previous works like LADE (Hong et al., 2021) and SADE (Zhang et al., 2022). The primary evaluation metric used is micro accuracy. To provide a comprehensive evaluation of the model's performance under varying test label distributions. Three types of test class distributions are typically considered: *i.e., uniform, forward* as mirrored in the training data, and *backward* where the class frequencies are reversed.

**Implementation details.** In our experiments, we adopt the NCL (Nested Collaborative Learning) (Li et al., 2022) as the base model by default. We follow the experimental setups in SADE for fair comparisons. In addition, we implement our *neural estimator* with a two-layer fully-connected neural network. Detailed configurations are elaborated in Appendix C.

## 4.2 COMPARISON WITH STATE-OF-THE-ART

Table 1 and Table 2 summarize the comparison results of LSC with existing methods on CIFAR100-LT, ImageNet-LT, and Places-LT. Across varying settings on three datasets, LSC consistently outperforms existing methods. Notably, our method particularly excels in tackling significant label distribution shifts, such as forward 50 and backward 50 scenarios. On the CIFAR100-LT dataset, LSC demonstrates substantial improvements over SADE, with performance boosts of 2.1 and 4.2 in these challenging scenarios. Similarly, on the ImageNet-LT dataset, LSC achieves 2.9 and 4.2 improvements. On the Places-LT dataset, the improvements are 1.3 and 2.8. For LADE which uses the ground-truth test label distribution and BalPoE which integrates complex data augmentations, LSC achieves even more significant performance improvement in most cases. As expected, existing methods assuming uniformly distributed test datasets lag far behind the performance of LSC when dealing with test-agnostic tasks.

Table 1: Test accuracy (%) on CIFAR-100-LT with imbalance factor 100 (ResNet32) and ImageNet-LT (ResNeXt50). *Prior*: test class distribution. ∗: Prior implicitly estimated from test data.

| | | CIFAR-100-LT-100 | | | | | ImageNet-LT | | | | |
| | | Forward | | Uni. | Backward | | Forward | | Uni. | Backward | |
| Method | Prior | 50 | 5 | 1 | 5 | 50 | 50 | 5 | 1 | 5 | 50 |
| Softmax | ✗ | 63.3 | 52.5 | 41.4 | 30.5 | 17.5 | 66.1 | 56.6 | 48.0 | 38.6 | 27.6 |
| MiSLAS | ✗ | 58.8 | 53.0 | 46.8 | 40.1 | 32.1 | 61.6 | 56.3 | 51.4 | 46.1 | 39.5 |
| LADE | ✗ | 56.0 | 51.0 | 45.6 | 40.0 | 34.0 | 63.4 | 57.4 | 52.3 | 46.8 | 40.7 |
| RIDE | ✗ | 63.0 | 53.6 | 48.0 | 38.1 | 29.2 | 67.6 | 61.7 | 56.3 | 51.0 | 44.0 |
| PaCo | ✗ | 62.0 | 57.6 | 52.2 | 47.0 | 40.7 | 66.6 | 62.7 | 58.9 | 54.1 | 48.7 |
| SADE | ✗ | 58.4 | 53.1 | 49.4 | 42.6 | 35.0 | 65.5 | 62.0 | 58.8 | 54.7 | 49.8 |
| BalPoE | ✗ | 65.1 | 54.8 | **52.0** | 44.6 | 36.1 | 67.6 | 63.3 | 59.8 | 55.7 | 50.8 |
| LADE | ✓ | 62.6 | 52.7 | 45.6 | 41.1 | 41.6 | 65.8 | 57.5 | 52.3 | 48.8 | 49.2 |
| SADE | ∗ | 65.9 | 54.8 | 49.8 | 44.7 | 42.4 | 69.4 | 63.0 | 58.8 | 55.5 | 53.1 |
| LSC (ours) | ∗ | **68.1** | **58.4** | 51.9 | **46.0** | **48.3** | **72.3** | **65.6** | **60.5** | **58.2** | **57.3** |

Table 2: Test accuracy (%) on Places-LT.

| | | Forward | | Uni. | Backward | |
| Method | Prior | 50 | 5 | 1 | 5 | 50 |
| Softmax | ✗ | 45.6 | 38.0 | 31.4 | 25.4 | 19.4 |
| *LA* | ✗ | 57.8 | 49.6 | 42.5 | 35.6 | 28.2 |
| MiSLAS | ✗ | 40.9 | 39.6 | 38.3 | 36.7 | 34.4 |
| LADE | ✗ | 42.8 | 40.8 | 39.2 | 37.6 | 35.7 |
| RIDE | ✗ | 43.1 | 42.0 | 40.3 | 38.7 | 36.9 |
| LADE | ✓ | 46.3 | 41.2 | 39.4 | 39.9 | 43.0 |
| SADE | ∗ | 46.4 | 42.6 | 40.9 | 41.1 | 41.6 |
| LSC | ∗ | **47.7** | **43.7** | **41.4** | **41.5** | **44.4** |

Table 3: Ablation studies on ImageNet-LT.

| | Forward | | Uni. | Backward | |
| Setting | 50 | 5 | 1 | 5 | 50 |
| Base model | 70.9 | 65.6 | 60.5 | 55.1 | 48.4 |
| model adjust | | | | | |
| w/ test prior | 72.8 | 65.4 | 60.5 | 57.6 | 57.5 |
| w/ DE | 71.7 | 65.5 | 60.0 | 54.7 | 48.7 |
| w/ LSC | 72.3 | 65.6 | 60.5 | 58.2 | 57.3 |
| logits clip | | | | | |
| w/o train clip | 72.2 | 65.6 | 60.0 | 54.4 | 48.2 |
| w/o test clip | 64.5 | 61.2 | 59.3 | 58.3 | 57.3 |
| w/o adaptive | 71.6 | 65.5 | 60.9 | 57.0 | 53.4 |

### 4.3 COMBINING LSC WITH EXISTING METHODS

In fact, LSC can be easily combined with many existing long-tail learning methods to cope with unknown test label distributions. Table 4 presents the results by combining LSC with RIDE (Wang et al., 2021b), PaCo (Cui et al., 2021), NCL (Li et al., 2022), and SHIKE (Jin et al., 2023). Note that we use NCL as the default base model in main results (*i.e.,* Table 1 and Table 2) and only train the model for 200 epochs which is consistent with SADE. In this experiment, NCL uses 400 training epochs as per the original paper. From the results, it can be seen that all methods exhibit performance improvements after combining with LSC. This implies that our method is not particularly designed for a particular model but is general enough to be readily applied to many existing methods. Furthermore, LSC consistently delivers more substantial performance enhancements when confronted with severe class imbalance in test data. Importantly, when the test label distribution is uniform, LSC is able to retain the performance of base models, avoiding incorrectly adjusting the model predictions.

### 4.4 IN-DEPTH ANALYSES

**Why does LSC improve the performance?**     The direct reason is that LSC can precisely estimate the test label distribution and adjust the model predictions to match the true distribution. Figures 1a to 1c depicts the estimated label distributions for three representative test label distributions, *i.e., forward*, *uniform*, and *backward*. The "Prior" is the true label distribution and the "Pre-adjusted" represents a direct estimate using base model predictions. The "Post-adjusted" is the model prediction after adjustment. Obviously, direct predictions by the base model are slightly biased towards tail classes in the case of *forward* long-tailed test data, whereas biased towards head classes in *uniform* and *backward* cases. In contrast, LSC can effectively adjust the model predictions and find a good balance between head and tail classes.

Table 4: Test accuracy (%) by combining LSC with existing methods.

| | CIFAR-100-LT-100 | | | | | ImageNet-LT | | | | |
| | Forward | | Uni. | Backward | | Forward | | Uni. | Backward | |
| Methods | 50 | 5 | 1 | 5 | 50 | 50 | 5 | 1 | 5 | 50 |
|---|---|---|---|---|---|---|---|---|---|---|
| RIDE | 64.1 | 55.9 | 48.6 | 40.8 | 31.5 | 66.4 | 60.8 | 55.7 | 50.1 | 44.0 |
| RIDE + LSC | **66.2** | **56.2** | 48.6 | **41.3** | **33.2** | **67.3** | **60.9** | 55.7 | **50.2** | **44.8** |
| Δ | +2.1 | +0.3 | +0.0 | +0.5 | +2.3 | +0.9 | +0.1 | +0.0 | +0.1 | +0.8 |
| PaCo | 62.0 | 57.6 | 52.2 | 47.0 | 40.7 | 66.6 | 62.7 | 58.9 | 54.1 | 48.7 |
| PaCo + LSC | **63.3** | **57.7** | 52.2 | **47.6** | **42.0** | **71.3** | 62.7 | 58.9 | **55.3** | **52.3** |
| Δ | +1.3 | +0.1 | +0.0 | +0.6 | +1.3 | +4.7 | +0.0 | +0.0 | +1.2 | +3.6 |
| NCL | 66.4 | 59.8 | 54.3 | 48.0 | 41.4 | 70.9 | 65.6 | 60.5 | 55.1 | 48.4 |
| NCL + LSC | **71.5** | **61.1** | 54.3 | **49.8** | **47.9** | **72.3** | 65.6 | 60.5 | **58.2** | **57.3** |
| Δ | +5.1 | +1.3 | +0.0 | +0.2 | +6.5 | +1.4 | +0.0 | +0.0 | +3.1 | +8.9 |
| SHIKE | 67.8 | 60.1 | 53.8 | 46.6 | 38.4 | - | - | - | - | - |
| SHIKE + LSC | **70.3** | **60.5** | 53.8 | **48.8** | **43.2** | - | - | - | - | - |
| Δ | +2.5 | +0.4 | +0.0 | +2.2 | +4.8 | - | - | - | - | - |

(a) *Forward*      (b) *Uniform*      (c) *Backward*      (d) *logit clipping*

Figure 1: (a-c) Model predictions before and after adjustment. (d) The demonstration of *logit clipping*. All experiments are conducted on ImageNet-LT.

**Why can LSC tackle varying test label distributions?** We attribute the success to the adaptive *logit clipping* which can choose the best value of $k$ automatically from the test data. Figure 1d illustrates the change of model performance as a function of $k$ on the ImageNet-LT dataset. As we can see, $k$ plays an important role in the method and can significantly impact the performance. As depicted, with an increase of the value of $k$, the accuracy consistently decreases in the *forward* case, while increases in the *backward* case. Fortunately, LSC is able to choose $k$ wisely and we denote the final choice of $k$ by the "star" for each test label distribution.

**Impact of each key component.** We conduct ablation studies on the core components of our method. Table 3 presents the results on the ImageNet-LT dataset with varying test label distributions. We first attempt to use a direct estimate (denoted as DE in the table) of test label distribution using the predictions of the base model. This brings a few performance improvements in comparison with the base model. If we assume that the test label distribution is accessible, it can be used to adjust the model outputs, which leads to substantial improvements. However, this assumption is usually invalid in practice. As an alternative, our method LSC can achieve comparable results without the need for the prior. Further, we study the effectiveness of the proposed adaptive logit clipping module. We sequentially remove logit clipping from LSC in training or test time, and replace the adaptive $k$ by a fixed value to see the influence. From the results, we can see that removing the training clip results in predictions biased towards the head class, whereas removing the test clip has opposite observations, *i.e.,* the prediction is biased towards the tail class. By removing adaptive $k$, the performance significantly deteriorates in the *backward-50* case.

**Why is fitting the unnormalized logits better than normalized probabilities?** In LSC, the average predicted logits over samples are used to train the *neural estimator*. One may ask if it is equivalent to using the normalized probabilities to achieve this. To make it more clear, given predicted logits $\widehat{Z}$ by the base model, the average logits for the $k$-th class is given as $\frac{1}{N}\sum_{i=1}^{N}\widehat{Z}_{ik}$ and the normalized version is $\frac{1}{N}\sum_{i=1}^{N}\widehat{P}_{ik}$ where $\widehat{P}_i = \text{softmax}(\widehat{Z}_i)$ for the $i$-th sample. First, Figure 2a shows the trend of average probabilities for classes. It can be seen that the discrepancy between

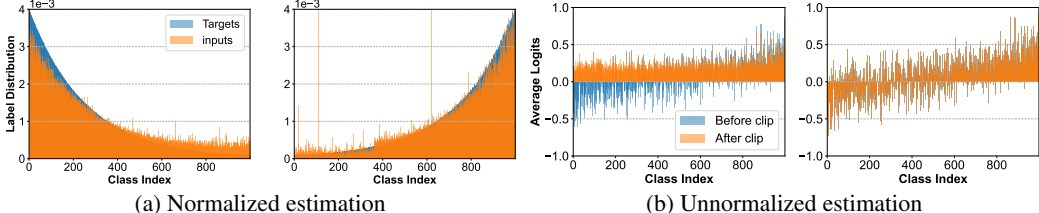

| (a) Normalized estimation | (b) Unnormalized estimation |

Figure 2: Comparison of using normalized and unnormalized average logits to train the *neural estimator*. Each panel has two plots, corresponding to the *Forward* and *Backward* settings.

Table 5: Test accuracy (%) on streaming test data with different batch sizes.

| | CIFAR-100-LT-100 | | | | | ImageNet-LT | | | | |
| | Forward | | Uni. | Backward | | Forward | | Uni. | Backward | |
| Methods (setting) | 50 | 5 | 1 | 5 | 50 | 50 | 5 | 1 | 5 | 50 |
| No test-time adaptation | 63.8 | 57.3 | 51.9 | 46.0 | 39.6 | 70.9 | 65.6 | 60.5 | 55.1 | 48.4 |
| Offline model (ours) | 68.1 | 58.4 | 51.9 | 46.0 | 48.3 | 72.3 | 65.6 | 60.5 | 58.2 | 57.3 |
| SADE (batch size 64) | 63.4 | 56.7 | 48.3 | 45.6 | 45.8 | 68.7 | 63.2 | 58.8 | 55.2 | 51.9 |
| Ours (batch size 64) | **68.7** | **61.4** | **51.3** | **47.2** | **48.0** | **71.8** | **65.7** | **60.8** | **56.5** | **52.8** |
| SADE (batch size 8) | 66.0 | 56.4 | 43.6 | 38.7 | 42.0 | 69.7 | 63.1 | 58.8 | 55.5 | **53.0** |
| Ours (batch size 8) | **68.8** | **57.1** | **51.1** | **46.9** | **48.0** | **71.8** | **65.7** | **60.8** | **56.5** | 52.8 |
| SADE (batch size 1) | 66.1 | 54.8 | 43.6 | 38.7 | 42.0 | 69.7 | 63.1 | 58.5 | 55.2 | **52.9** |
| Ours (batch size 1) | **68.4** | **56.4** | **51.7** | **46.1** | **48.1** | **71.9** | **65.7** | **60.7** | **56.1** | 52.3 |

inputs and targets for training *neural estimator* are marginal, in which case the neural network only needs to learn an identical mapping to achieve a small training error. However, this approach fails to generalize to the test data on which the base model has unsatisfactory performance as depicted in Figures 1a to 1c. In contrast, LSC opts to use average logits as the input. Since the pre-trained model tends to suppress the head class predictions (*i.e.,* many head class logits are negative), the average logits do not exhibit obvious differences across varying test label distributions. This makes *neural estimator* hard to correlate the inputs with targets. By applying logit clipping, it mitigates this issue, as shown in Figure 2b. We suspect that fitting normalized probabilities may work well if an extra validation set is accessible. This is because the pre-trained model has already overfit the training set and the predicted probabilities are not informative enough to distinguish the label distribution.

### 4.5 EVALUATING LSC UNDER THE ONLINE SETTING

We also test our method in the online setting rather than assuming that all test data is accessible in advance. In this setup, we estimate the overall test distribution using the exponential moving average which accumulates over each batch of test data points. Table 5 presents the results in comparison with SADE by setting different sizes of mini-batch. Specifically, the test data comes one by one when setting the batch size to 1. From the results, it can be seen that LSC performs well on streaming test data and surpasses SADE in most cases. Moreover, LSC achieves comparable performance with its offline counterpart, which validates the robustness of the test label distribution estimation approach.

## 5 CONCLUSION

In summary, this paper introduces a straightforward yet effective approach for addressing test-agnostic long-tail learning. Our method leverages a shallow neural network equipped with an adaptive logit clipping module to estimate the true test label distribution. Extensive experiments on CIFAR100-LT, ImageNet-LT, and Places-LT demonstrate that our method consistently outperforms existing state-of-the-art methods, even when confronted with varying test label distributions. Furthermore, our method is versatile and can be seamlessly integrated with numerous existing long-tail learning models to enhance their generalization capabilities in scenarios with unknown test label distributions. We hope that our work can shed light on future research aimed at addressing test label shifts.

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

## A    PROOF OF THEOREM 3.1

*Proof.* To upper bound the error gap between the adjusted classifier and the Bayes-optimal classifier, we first rewrite the Bayes-optimal classifier $h^*$ as follows:

$$h^*(x) = \arg\max_{y \in [K]} \frac{\mathbb{P}_{\mathcal{D}_T}(Y = y)}{\mathbb{P}_{\mathcal{D}_S}(Y = y)} \mathbb{P}_{\mathcal{D}_S}(Y = y \mid x)$$

For simplicity, we denote the prediction of the base model as $\widehat{Y} = h_f(x)$. By definition, the Bayes error can be written as:

$$\epsilon_T(h^*) = \min_{\widehat{Y}} \sum_{y \in [K]} \mathbb{P}_{\mathcal{D}_T}(\widehat{Y} \neq y \mid Y = y) \mathbb{P}_{\mathcal{D}_T}(Y = y)$$

$$= \min_{\widehat{Y}} 1 - \sum_{y \in [K]} \int_{x \in \mathcal{X}} \mathbb{P}_{\mathcal{D}_T}(\widehat{Y} = y \mid x) \mathbb{P}_{\mathcal{D}_T}(x \mid Y = y) \mathbb{P}_{\mathcal{D}_T}(Y = y) \, \mathrm{d}x$$

$$= \min_{\widehat{Y}} 1 - \sum_{y \in [K]} \int_{x \in \mathcal{X}} \mathbb{P}_{\mathcal{D}_T}(\widehat{Y} = y \mid x) \mathbb{P}_{\mathcal{D}_S}(x \mid Y = y) \mathbb{P}_{\mathcal{D}_T}(Y = y) \, \mathrm{d}x$$

$$= \min_{\widehat{Y}} 1 - \sum_{y \in [K]} \int_{x \in \mathcal{X}} \mathbb{P}_{\mathcal{D}_T}(\widehat{Y} = y \mid x) \frac{\mathbb{P}_{\mathcal{D}_T}(Y = y) \mathbb{P}_{\mathcal{D}_S}(Y = y \mid x)}{\mathbb{P}_{\mathcal{D}_S}(Y = y)} \mathbb{P}_{\mathcal{D}_S}(X = x) \, \mathrm{d}x$$

$$= 1 - \max_{\widehat{Y}} \int_x \sum_{y \in [K]} \left( \mathbb{P}_{\mathcal{D}_T}(\widehat{Y} = y \mid x) \frac{\mathbb{P}_{\mathcal{D}_T}(Y = y) \mathbb{P}_{\mathcal{D}_S}(Y = y \mid x)}{\mathbb{P}_{\mathcal{D}_S}(Y = y)} \right) \mathbb{P}_{\mathcal{D}_S}(X = x) \, \mathrm{d}x$$

It can be seen that minimizing the Bayes error of test domain $\epsilon_T(h^*)$ is equivalent to maximizing $\sum_{y \in [K]} \left( \mathbb{P}_{\mathcal{D}_T}(\widehat{Y} = y \mid x) \frac{\mathbb{P}_{\mathcal{D}_T}(Y=y) \mathbb{P}_{\mathcal{D}_S}(Y=y \mid x)}{\mathbb{P}_{\mathcal{D}_S}(Y=y)} \right)$ for each $x \in \mathcal{X}$. Since $\sum_{y \in [K]} \mathbb{P}_{\mathcal{D}_T}(\widehat{Y} = y \mid x) = 1$, the Bayes-optimal decision function is as follows:

$$\mathbb{P}_{\mathcal{D}_T}(\widehat{Y} = y \mid x) = \begin{cases} 1, & y = \arg\max_{y \in [K]} \frac{\mathbb{P}_{\mathcal{D}_T}(Y=y)}{\mathbb{P}_{\mathcal{D}_S}(Y=y)} \mathbb{P}_{\mathcal{D}_S}(Y = y \mid x) \\ 0, & \text{otherwise} \end{cases}$$

We rewrite $\widehat{Y} = \arg\max_{y \in [K]} \frac{\widehat{\mathbb{P}}_{\mathcal{D}_T}(Y=y)}{\mathbb{P}_{\mathcal{D}_S}(Y=y)} \widehat{\mathbb{P}}_{\mathcal{D}_S}(Y = y \mid x)$ and denote $\widehat{Y}^* = h^*(x)$. To simplify the notations, we define $\mathbb{P}_{S,T}(\cdot) = \frac{\mathbb{P}_{\mathcal{D}_T}(Y=\cdot)}{\mathbb{P}_{\mathcal{D}_S}(Y=\cdot)} \mathbb{P}_{\mathcal{D}_S}(Y = \cdot \mid x)$ and $\widehat{\mathbb{P}}_{S,T}(\cdot) = \frac{\widehat{\mathbb{P}}_{\mathcal{D}_T}(Y=\cdot)}{\mathbb{P}_{\mathcal{D}_S}(Y=\cdot)} \widehat{\mathbb{P}}_{\mathcal{D}_S}(Y = \cdot \mid x)$. We now write the error gap between them as follows:

$$\epsilon_T(\widehat{Y}) - \epsilon_T(\widehat{Y}^*) = \int_{x \in \widehat{Y} \neq \widehat{Y}^*} \left( \mathbb{P}_{S,T}(\widehat{Y}^*) - \mathbb{P}_{S,T}(\widehat{Y}) \right) \mathbb{P}_{\mathcal{D}_S}(X = x) \, \mathrm{d}x$$

$$= \int_{x \in \widehat{Y} \neq \widehat{Y}^*} \left[ \left( \widehat{\mathbb{P}}_{S,T}(\widehat{Y}) - \mathbb{P}_{S,T}(\widehat{Y}) \right) + \left( \mathbb{P}_{S,T}(\widehat{Y}^*) - \widehat{\mathbb{P}}_{S,T}(\widehat{Y}^*) \right) \right] \mathbb{P}_{\mathcal{D}_S}(X = x) \, \mathrm{d}x$$

$$+ \int_{x \in \widehat{Y} \neq \widehat{Y}^*} \left( \widehat{\mathbb{P}}_{S,T}(\widehat{Y}^*) - \widehat{\mathbb{P}}_{S,T}(\widehat{Y}) \right) \mathbb{P}_{\mathcal{D}_S}(X = x) \, \mathrm{d}x$$

$$\leq \int_{x \in \widehat{Y} \neq \widehat{Y}^*} \left[ \sum_{y \in [K]} \left| \widehat{\mathbb{P}}_{S,T}(y) - \mathbb{P}_{S,T}(y) \right| \right] \mathbb{P}_{\mathcal{D}_S}(X = x) \, \mathrm{d}x$$

Here, the term $\left| \widehat{\mathbb{P}}_{S,T}(y) - \mathbb{P}_{S,T}(y) \right|$ can be bounded as follows:

$$\left| \widehat{\mathbb{P}}_{S,T}(y) - \mathbb{P}_{S,T}(y) \right| \leq \left| \frac{\widehat{\mathbb{P}}_{\mathcal{D}_S}(Y = y \mid x) \widehat{\mathbb{P}}_{\mathcal{D}_T}(Y = y)}{\mathbb{P}_{\mathcal{D}_S}(Y = y)} - \frac{\mathbb{P}_{\mathcal{D}_S}(Y = y \mid x) \widehat{\mathbb{P}}_{\mathcal{D}_T}(Y = y)}{\mathbb{P}_{\mathcal{D}_S}(Y = y)} \right|$$

$$+ \frac{\mathbb{P}_{\mathcal{D}_S}(Y = y \mid x)}{\mathbb{P}_{\mathcal{D}_S}(Y = y)} \left| \mathbb{P}_{\mathcal{D}_T}(Y = y) - \widehat{\mathbb{P}}_{\mathcal{D}_T}(Y = y) \right|$$

Substitute it into the original bounds, we have:

$$
\epsilon_T(\widehat{Y}) - \epsilon_T \leq \int_{x \in \widehat{Y} \neq \widehat{Y}^*} \left[ \sum_{y \in [K]} \left| \widehat{\mathbb{P}}_{S,T}(y) - \mathbb{P}_{S,T}(y) \right| \right] \mathbb{P}_{\mathcal{D}_S}(X = x) \, \mathrm{d}x
$$

$$
\leq \int_{x \in \widehat{Y} \neq \widehat{Y}^*} \left[ \sum_{y \in [K]} w_y \left| \widehat{\mathbb{P}}_{\mathcal{D}_S}(y \mid x) - \mathbb{P}_{\mathcal{D}_S}(y \mid x) \right| \right] \mathbb{P}_{\mathcal{D}_S}(X = x) \, \mathrm{d}x
$$

$$
+ \sum_{y \in [K]} \left| \mathbb{P}_{\mathcal{D}_T}(Y = y) - \widehat{\mathbb{P}}_{\mathcal{D}_T}(Y = y) \right| \int_{x \in \widehat{Y} \neq \widehat{Y}^*} \mathbb{P}_{\mathcal{D}_S}(x \mid Y = y) \, \mathrm{d}x
$$

By definition, it can be directly obtained that the first term is upper bounded by:

$$
\mathbb{E}_{x \sim \mathcal{D}_S} \left[ \left\| \widehat{\mathbb{P}}(Y \mid x) - \mathbb{P}_{\mathcal{D}_S}(Y \mid x) \right\|_{L^1, w} \right] = \left\| \widehat{\mathbb{P}}(Y \mid X) - \mathbb{P}_{\mathcal{D}_S}(Y \mid X) \right\|_{L^1, w}
$$

Moreover, the second term has an upper bound as follows:

$$
\left\| \widehat{\mathbb{P}}_{\mathcal{D}_T}(Y) - \mathbb{P}_{\mathcal{D}_T}(Y) \right\|_{L^1} \max_{y \in [K]} \int_{x \in \widehat{Y} \neq \widehat{Y}^*} \mathbb{P}_{\mathcal{D}_S}(x \mid Y = y) \, \mathrm{d}x
$$

Finally, it can be confirmed by directly computing that $\max_{y \in [K]} \int_{x \in \widehat{Y} \neq \widehat{Y}^*} \mathbb{P}_{\mathcal{D}_S}(x \mid Y = y) \, \mathrm{d}x$ equals to $\mathrm{BSE}(h)$. Combining all the conclusions above ends the proof. $\qquad \square$

## B    PROOF OF THEOREM 3.2

*Proof.* For simplicity, we denote $C = \widehat{C}_{h_f(X) \mid Y}$. We know that $\Pi' = \{C\pi \mid \pi \in \Pi\}$ is an invertible condition confusion matrix because $\sum_{k=1}^{K} C_{kj} = 1, j \in [K], \Pi' \in \Pi$. Moreover, we only consider joint distribution $\mathcal{D}_L$ and denote $\mathbb{P}_{\mathcal{D}_L}(\cdot)$ as $\mathbb{P}(\cdot)$ in the following. We first prove the approximation error measured by $L^2$ distance:

$$
\epsilon(g^*) = \min_g \mathbb{E}_{(\pi, \widetilde{\pi}) \sim \mathcal{D}} \left[ \| \pi - g(\widetilde{\pi}) \|_{L^2}^2 \right] = \mathbb{E}_{\widetilde{\pi}} \left[ \mathrm{Var}\left[ \pi \mid \widetilde{\pi} \right] \right]
$$

In order to bound $\epsilon(g^*)$, we approximate $\mathrm{Var}[\pi \mid \widetilde{\pi}]$ as follows:

$$
\mathrm{Var}[\pi \mid \widetilde{\pi}] = \int_{\pi \in \Pi} \| \pi - \mathbb{E}[\pi \mid \widetilde{\pi}] \|_{L^2}^2 \, \mathbb{P}(\pi \mid \widetilde{\pi}) \, \mathrm{d}\pi
$$

$$
= \int_{\pi \in \Pi} \left\| C^{-1} \left( C\pi - C\mathbb{E}[\pi \mid \widetilde{\pi}] \right) \right\|_{L^2}^2 \, \mathbb{P}(\pi \mid \widetilde{\pi}) \, \mathrm{d}\pi
$$

$$
\leq \left\| C^{-1} \right\|_{L^2}^2 \mathrm{Var}[C\pi \mid \widetilde{\pi}]
$$

Here, $\left\| C^{-1} \right\|_{L^2}$ denotes the spectral norm of matrix $C$, and the inequality holds because $\left\| C^{-1} \left( C\pi - C\mathbb{E}[\pi \mid \widetilde{\pi}] \right) \right\|_{L^2}^2 \leq \left\| C^{-1} \right\|_{L^2}^2 \left\| C\pi - \mathbb{E}[C\pi \mid \widetilde{\pi}] \right\|_{L^2}^2$.

We now analyze the posterior distribution of $C\pi$ given pseudo label distribution $\widetilde{\pi}$. We denote $\pi' = C\pi$. As the marginal prior label distribution is a uniform distribution on label space $\Pi$, the distribution of $\pi'$ is also a uniform distribution on $\Pi'$. Moreover, we use $\mathrm{Dirichlet}(\cdot)$ denote *Dirichlet distribution*, hence we have $\pi \sim \mathrm{Dirichlet}(1, ..., 1)$. Then, the probability density function of $\pi'$ is:

$$
\mathbb{P}(\pi') = \begin{cases} \frac{1}{|\det(C)|} \mathbb{P}(\pi) = \frac{1}{|\det(C)|} \Gamma(K), & \pi' \in \Pi' \\ 0, & \text{otherwise} \end{cases}
$$

Hence, we can expand $\mathrm{Var}[C\pi \mid \widetilde{\pi}]$ through using Bayesian formula to calculate $\mathbb{P}(\pi' \mid \widetilde{\pi})$:

$$
\begin{aligned}
\mathrm{Var}[C\pi \mid \widetilde{\pi}] &= \int_{\pi' \in \Pi'} \|\pi' - \mathbb{E}[\pi' \mid \widetilde{\pi}]\|_{L^2}^2 \, \mathbb{P}(\pi' \mid \widetilde{\pi}) \, \mathrm{d}\pi' \\
&= \int_{\pi' \in \Pi'} \frac{\|\pi' - \mathbb{E}[\pi' \mid \widetilde{\pi}]\|_{L^2}^2 \, \mathbb{P}(\widetilde{\pi} \mid \pi')\Gamma(k)}{|\det(C)|\mathbb{P}(\widetilde{\pi})} \, \mathrm{d}\pi' \\
&= \frac{\Gamma(k)}{\mathbb{P}(\widetilde{\pi})\,|\det(C)|} \int_{\pi' \in \Pi'} \|\pi' - \mathbb{E}[\pi' \mid \widetilde{\pi}]\|_{L^2}^2 \, \frac{\Gamma(M+1)}{\prod_{k=1}^{K}\Gamma(M_k+1)} \prod_{k=1}^{K}(\pi'_k)^{M_k} \, \mathrm{d}\pi' \\
&= \frac{\Gamma(M+1)\Gamma(K)}{\Gamma(M+K)\mathbb{P}(\widetilde{\pi})\,|\det(C)|} \int_{\pi' \in \Pi'} \|\pi' - \mathbb{E}[\pi' \mid \widetilde{\pi}]\|_{L^2}^2 \, \frac{\Gamma(M+K)}{\prod_{k=1}^{K}\Gamma(M_k+1)} \prod_{k=1}^{K}(\pi'_k)^{M_k} \, \mathrm{d}\pi'
\end{aligned}
$$

Since directly calculating $\mathbb{E}[\pi' \mid \widetilde{\pi}]$ is infeasible, we consider $\widehat{\mathbb{E}}[\pi' \mid \widetilde{\pi}]$ as follows:

$$
\begin{aligned}
\widehat{\mathbb{E}}[\pi' \mid \widetilde{\pi}] &= \int_{\pi' \in \Pi} \pi' \frac{\Gamma(M+K)}{\prod_{k=1}^{K}\Gamma(M_k+1)} \prod_{k=1}^{K}(\pi'_k)^{M_k} \\
&= \left( \frac{M_1+1}{M+K}, \frac{M_2+1}{M+K}, \cdots, \frac{M_K+1}{M+K} \right)^{\top}
\end{aligned}
$$

Then, we can bound $\mathrm{Var}[C\pi \mid \widetilde{\pi}]$ as follows:

$$
\begin{aligned}
\mathrm{Var}&[C\pi \mid \widetilde{\pi}] \\
&\leq \frac{\Gamma(M+1)\Gamma(K)}{\Gamma(M+K)\mathbb{P}(\widetilde{\pi})\,|\det(C)|} \int_{\pi' \in \Pi'} \left\| \pi' - \widehat{\mathbb{E}}[\pi' \mid \widetilde{\pi}] \right\|_{L^2}^2 \frac{\Gamma(M+K)}{\prod_{k=1}^{K}\Gamma(M_k+1)} \prod_{k=1}^{K}(\pi'_k)^{M_k} \, \mathrm{d}\pi' \\
&\leq \frac{\Gamma(M+1)\Gamma(K)}{\Gamma(M+K)\mathbb{P}(\widetilde{\pi})\,|\det(C)|} \int_{\pi' \in \Pi} \left\| \pi' - \widehat{\mathbb{E}}[\pi' \mid \widetilde{\pi}] \right\|_{L^2}^2 \frac{\Gamma(M+K)}{\prod_{k=1}^{K}\Gamma(M_k+1)} \prod_{k=1}^{K}(\pi'_k)^{M_k} \, \mathrm{d}\pi'
\end{aligned}
$$

The first inequality holds because for any random variable $\pi'$ and any vector $w$, we have $\mathbb{E}\left[\|\pi' - \mathbb{E}[\pi']\|_{L^2}^2\right] \leq \mathbb{E}\left[\|\pi' - w\|_{L^2}^2\right]$. The second inequality holds simply because $\Pi' \subset \Pi$. Indeed, $\frac{\Gamma(M+K)}{\prod_{k=1}^{K}\Gamma(M_k+1)} \prod_{k=1}^{K}(\pi'_k)^{M_k}$ is the probability density function of $\mathrm{Dirichlet}(M_1 + 1, \cdots, M_K + 1)$. Notably, $\widehat{\mathbb{E}}[\pi' \mid \widetilde{\pi}]$ equals to the expectations of distribution $\mathrm{Dirichlet}(M_1 + 1, \cdots, M_K + 1)$. Therefore, we have:

$$
\begin{aligned}
\int_{\pi' \in \Pi} \left\| \pi' - \widehat{\mathbb{E}}[\pi' \mid \widetilde{\pi}] \right\|_{L^2}^2 &\frac{\Gamma(M+K)}{\prod_{k=1}^{K}\Gamma(M_k+1)} \prod_{k=1}^{K}(\pi'_k)^{M_k} \, \mathrm{d}\pi' \\
&= \mathrm{Tr}\left[\mathrm{Cov}(\mathrm{Dirichlet}(M_1 + 1, \cdots, M_K + 1)\right] \\
&= \frac{(M+K)^2 - \sum_{k=1}^{K}(M_k+1)^2}{(M+K)^2(M+K+1)}
\end{aligned}
$$

On this basis, we can bound $\epsilon_{g^*}$ as follows:

$$
\begin{aligned}
\epsilon(g^*) &= \mathbb{E}_{\widetilde{\pi}}\left[\mathrm{Var}\left[\pi \mid \widetilde{\pi}\right]\right] = \sum_{\widetilde{\pi} \in \widetilde{\Pi}} \mathrm{Var}[\pi \mid \widetilde{\pi}]\mathbb{P}(\widetilde{\pi}) \\
&\leq \sum_{\widetilde{\pi} \in \widetilde{\Pi}} \left\|C^{-1}\right\|^2 \mathrm{Var}[\pi' \mid \widetilde{\pi}]\mathbb{P}(\widetilde{\pi}) \\
&= \frac{\left\|C^{-1}\right\|^2}{|\det(C)|} \sum_{\widetilde{\pi} \in \widetilde{\Pi}} \frac{\Gamma(M+1)\Gamma(K)}{\Gamma(M+K)} \frac{(M+K)^2 - \sum_{k=1}^{K}(M_k+1)^2}{(M+K)^2(M+K+1)} \\
&\leq \frac{\left\|C^{-1}\right\|^2}{|\det(C)|} \left( \max_{\widetilde{\pi} \in \widetilde{\Pi}} \frac{(M+K)^2 - \sum_{k=1}^{K}(M_k+1)^2}{(M+K)^2(M+K+1)} \right) \sum_{\widetilde{\pi} \in \widetilde{\Pi}} \frac{\Gamma(M+1)\Gamma(K)}{\Gamma(M+K)}
\end{aligned}
$$

Noting that the maximum value of $\frac{(M+K)^2 - \sum_{k=1}^{K}(M_k+1)^2}{(M+K)^2(M+K+1)}$ is $\frac{K-1}{K(M+K+1)}$ when $M_K + 1 = \frac{M+K}{K}$. Moreover, we have $\sum_{\widetilde{\pi} \in \widetilde{\Pi}} \frac{\Gamma(M+1)\Gamma(K)}{\Gamma(M+K)} = 1$. Let $\sigma_{min}$ denote minimum eigenvalue of the $C$, we prove the following upper bound:

$$\epsilon(g^*) \leq \frac{K-1}{|\det(C)|\,\sigma_{min}^2(M+K+1)K}$$

Additionally, it is worth noting that lower bounds can be directly calculated when the pseudo-label matches the true label, *i.e.,* $C = I$ (the identity matrix). In such cases, the inequations in the proof can be replaced by equations. Building upon this observation, we can conclude the proof. □

## C ADDITIONAL EXPERIMENTAL DETAILS

### C.1 DATASET DETAILS

In this section, we provide additional details about the datasets used in our experiments. The paper focuses on three widely used datasets: ImageNet-LT, Places-LT, and CIFAR100-LT. Below, we summarize the key characteristics of these datasets, as presented in Table 6.

Table 6: Statistics of datasets.

| Dataset | # Classes | # Training data | # Test data | Imbalance ratio |
|---|---|---|---|---|
| ImageNet-LT (Liu et al., 2019) | 1,000 | 115,846 | 50,000 | 256 |
| Places-LT (Liu et al., 2019) | 365 | 62,500 | 36,500 | 996 |
| CIFAR100-LT (Cao et al., 2019) | 100 | 50,000 | 10,000 | {50,100} |

### C.2 IMPLEMENTATION DETAILS

By default, we use NCL as the base model, removing its contrastive learning module. For model training, we employ SGD with momentum as the optimization algorithm across all datasets. On CIFAR-100, we utilize ResNet32 (He et al., 2016) as the backbone and train for 200 epochs. The batch size is set to 128, the initial learning rate is 0.01, and the weight decay factor is $2e^{-4}$. On ImageNet-LT, we choose ResNeXt50 (Xie et al., 2017) as the backbone and train for 180 epochs. The batch size is set to 64, the initial learning rate is 0.0025, and the weight decay factor is $5e^{-4}$. On Places-LT, we use a pre-trained ResNet152 as the backbone and train for 30 epochs. The batch size is set to 128, the initial learning rate is 0.01, and the weight decay factor is $4e^{-4}$. We maintain identical experimental setups to SADE to ensure fair and consistent comparisons. To train our *neural estimator*, we systematically construct "datasets" to cover a range of imbalance degrees, varying from Forward-100 to Backward-100, with an interval of 0.1. Throughout the training process, we apply different logit clipping ratios depending on the dataset: 0.8 for CIFAR100-LT, 0.99 for ImageNet-LT, and 0.95 for Places-LT. We employ the KL divergence as the loss function and use the Adam optimizer to train the model for 100 epochs. The learning rates are configured as follows: $1e^{-3}$ for CIFAR100-LT, $1e^{-5}$ for ImageNet-LT, and $1e^{-5}$ for Places-LT. At test time, the final value of $k$ is adaptively chosen from the set $\{0.1K, 0.2K, \cdots, 0.9K, K\}$ where $K$ is the total number of classes.

### C.3 FULL RESULTS UNDER VARYING TEST LABEL DISTRIBUTIONS

We conduct additional experiments to showcase the effectiveness of our method across a wider range of imbalance ratios in both the training dataset and test label distributions. Detailed results for ImageNet-LT, Places-LT, CIFAR100-LT-100, and CIFAR100-LT-50 datasets can be found in Tables 7 to 10. These results clearly demonstrate that LSC consistently outperforms previous state-of-the-art methods such as SADE and BalPoE. In addition, we compare with methods: 1) test class distribution estimation leverages BBSE (Lipton et al., 2018) and 2) Tent which fine-tunes the batch normalization layers via entropy minimization on test data (Wang et al., 2021a). Both methods are built upon

Table 7: Test accuracy (%) on multiple test distributions for ResNeXt50 trained on ImageNet-LT. *Prior*: test class distribution is used. ∗: Prior implicitly estimated from test data.

| Method | Prior | Forward | | | | Uni. | Backward | | | |
| --- | --- | --- | --- | --- | --- | --- | --- | --- | --- | --- |
| | | 50 | 25 | 10 | 5 | 1 | 5 | 10 | 25 | 50 |
| Softmax | ✗ | 66.1 | 63.8 | 60.3 | 56.6 | 48.0 | 38.6 | 34.9 | 30.9 | 27.6 |
| LA | ✗ | 63.2 | 61.9 | 59.5 | 57.2 | 52.3 | 47.0 | 45.0 | 42.3 | 40.8 |
| MiSLAS | ✗ | 61.6 | 60.4 | 58.0 | 56.3 | 51.4 | 46.1 | 44.0 | 41.5 | 39.5 |
| LADE | ✗ | 63.4 | 62.1 | 59.9 | 57.4 | 52.3 | 46.8 | 44.9 | 42.7 | 40.7 |
| RIDE | ✗ | 67.6 | 66.3 | 64.0 | 61.7 | 56.3 | 51.0 | 48.7 | 46.2 | 44.0 |
| SADE | ✗ | 65.5 | 64.4 | 63.6 | 62.0 | 58.8 | 54.7 | 53.1 | 51.1 | 49.8 |
| BalPoE | ✗ | 67.6 | 66.3 | 65.2 | 63.3 | 59.8 | 55.7 | 54.3 | 52.2 | 50.8 |
| NCL | ✗ | 70.9 | 69.9 | 67.8 | 65.6 | 60.5 | 55.1 | 52.4 | 50.5 | 48.4 |
| LADE | ✓ | 65.8 | 63.8 | 60.6 | 57.5 | 52.3 | 48.8 | 48.6 | 49.0 | 49.2 |
| SADE | ∗ | 69.4 | 67.4 | 65.4 | 63.0 | 58.8 | 55.5 | 54.5 | 53.7 | 53.1 |
| SADE + BBSE | ∗ | 69.1 | 66.6 | 63.7 | 60.5 | 53.3 | 45.6 | 42.7 | 39.5 | 36.8 |
| SADE + Tent | ∗ | 68.0 | 67.0 | 65.6 | 62.8 | 58.6 | 53.2 | 50.6 | 48.1 | 45.7 |
| LSC | ∗ | **72.3** | **70.8** | **68.1** | **65.6** | **60.5** | **58.3** | **57.8** | **57.9** | **57.3** |

Table 8: Test accuracy (%) on multiple test distributions for ResNet152 trained on Places-LT. *Prior*: test class distribution is used. ∗: Prior implicitly estimated from test data.

| Method | Prior | Forward | | | | Uni. | Backward | | | |
| --- | --- | --- | --- | --- | --- | --- | --- | --- | --- | --- |
| | | 50 | 25 | 10 | 5 | 1 | 5 | 10 | 25 | 50 |
| Softmax | ✗ | 45.6 | 42.7 | 40.2 | 38.0 | 31.4 | 25.4 | 23.4 | 20.8 | 19.4 |
| LA | ✗ | 42.7 | 41.7 | 41.3 | 41.0 | 39.4 | 37.8 | 37.1 | 36.2 | 35.6 |
| MiSLAS | ✗ | 40.9 | 39.7 | 39.5 | 39.6 | 38.3 | 36.7 | 35.8 | 34.7 | 34.4 |
| LADE | ✗ | 42.8 | 41.5 | 41.2 | 40.8 | 39.2 | 37.6 | 36.9 | 36.0 | 35.7 |
| RIDE | ✗ | 43.1 | 41.8 | 41.6 | 42.0 | 40.3 | 38.7 | 38.2 | 37.0 | 36.9 |
| LADE | ✓ | 46.3 | 44.2 | 42.2 | 41.2 | 39.4 | 39.9 | 40.9 | 42.4 | 43.0 |
| SADE | ∗ | 46.4 | 44.9 | 43.3 | 42.6 | 40.9 | 41.1 | 41.4 | 42.0 | 41.6 |
| LSC | ∗ | **48.7** | **46.5** | **44.8** | **43.8** | **41.4** | **41.6** | **44.8** | **43.0** | **44.5** |

SADE. It can be seen that incorporating BBSE and Tent leads to significant performance drops in the *backward* scenarios, particularly for BBSE. This showcases the robustness and superior performance of our method when faced with varying test label distributions.

Furthermore, we investigate how our proposed method impacts the performance of both head and tail classes. In Table 11, we present the accuracy for three splits of classes: many-shot classes ($>$100 images), medium-shot classes (20∼100 images), and few-shot classes ($<$20 images). The results highlight the adaptability of LSC in improving the performance of either head or tail classes based on the specific test label distribution. Notably, when the test label distribution follows a "forward" long-tail distribution, LSC demonstrates more significant performance improvements on head classes. Conversely, when the test label distribution conforms to a "backward" long-tail distribution, LSC exhibits more pronounced enhancements in the performance of tail classes. The results underscore the effectiveness of our approach in addressing test-agnostic long-tailed label distributions.

### C.4 IMPACT OF NEURAL ESTIMATOR ARCHITECTURE

In this experiment, we explore the effects of varying the number of layers in our proposed *neural estimator*. The results, as shown in Table 12, highlight the influence of different layer configurations on model performance. Surprisingly, a single-layer neural estimator demonstrates a noteworthy improvement in performance compared to the base model. Additionally, employing multiple layers

Table 9: Test accuracy (%) on multiple test distributions for model trained on CIFAR-100-LT-100. *Prior*: test class distribution is used. ∗: Prior implicitly estimated from test data.

| Method | Prior | Forward | | | | Uni. | Backward | | | |
| | | 50 | 25 | 10 | 5 | 1 | 5 | 10 | 25 | 50 |
|---|---|---|---|---|---|---|---|---|---|---|
| Softmax | ✗ | 63.3 | 62.0 | 56.2 | 52.5 | 41.4 | 30.5 | 25.8 | 21.7 | 17.5 |
| LA | ✗ | 57.8 | 55.5 | 54.2 | 52.0 | 46.1 | 40.8 | 38.4 | 36.3 | 33.7 |
| MiSLAS | ✗ | 58.8 | 57.2 | 55.2 | 53.0 | 46.8 | 40.1 | 37.7 | 33.9 | 32.1 |
| LADE | ✗ | 56.0 | 55.5 | 52.8 | 51.0 | 45.6 | 40.0 | 38.3 | 35.5 | 34.0 |
| RIDE | ✗ | 63.0 | 59.9 | 57.0 | 53.6 | 48.0 | 38.1 | 35.4 | 31.6 | 29.2 |
| SADE | ✗ | 58.4 | 57.0 | 54.4 | 53.1 | 49.4 | 42.6 | 39.7 | 36.7 | 35.0 |
| BalPoE | ✗ | 65.1 | 63.1 | 60.8 | **58.4** | **52.0** | 44.6 | 41.8 | 38.0 | 36.1 |
| NCL | ✗ | 63.8 | 62.4 | 59.6 | 57.3 | 51.9 | 46.0 | 43.6 | 40.9 | 39.6 |
| LADE | ✓ | 62.6 | 60.2 | 55.6 | 52.7 | 45.6 | 41.1 | 41.5 | 40.7 | 41.6 |
| SADE | ∗ | 65.9 | 62.5 | 58.3 | 54.8 | 49.8 | 44.7 | 43.9 | 42.5 | 42.4 |
| LSC | ∗ | **68.1** | **65.6** | **61.7** | **58.4** | 51.9 | **46.0** | **47.9** | **47.4** | **48.3** |

Table 10: Test accuracy (%) on multiple test distributions for model trained on CIFAR-100-LT-50. *Prior*: test class distribution is used. ∗: Prior implicitly estimated from test data.

| Method | Prior | Forward | | | | Uni. | Backward | | | |
| | | 50 | 25 | 10 | 5 | 1 | 5 | 10 | 25 | 50 |
|---|---|---|---|---|---|---|---|---|---|---|
| Softmax | ✗ | 64.8 | 62.7 | 58.5 | 55.0 | 45.6 | 36.2 | 32.1 | 26.6 | 24.6 |
| LA | ✗ | 61.6 | 60.2 | 58.4 | 55.9 | 50.9 | 45.7 | 43.9 | 42.5 | 40.6 |
| MiSLAS | ✗ | 60.1 | 58.9 | 57.7 | 56.2 | 51.5 | 46.5 | 44.3 | 41.8 | 40.2 |
| LADE | ✗ | 61.3 | 60.2 | 56.9 | 54.3 | 50.1 | 45.7 | 44.0 | 41.8 | 40.5 |
| RIDE | ✗ | 62.2 | 61.0 | 58.8 | 56.4 | 51.7 | 44.0 | 41.4 | 38.7 | 37.1 |
| SADE | ✗ | 59.5 | 58.6 | 56.4 | 54.8 | 53.8 | 48.2 | 46.1 | 44.4 | 43.6 |
| BalPoE | ✗ | 66.5 | 64.8 | **62.8** | **60.9** | **56.3** | **51.0** | 48.9 | 46.6 | 45.3 |
| NCL | ✗ | 63.2 | 61.8 | 60.1 | 58.5 | 53.4 | 48.6 | 46.8 | 44.0 | 43.1 |
| LADE | ✓ | 65.9 | 62.1 | 58.8 | 56.0 | 50.1 | 45.5 | 46.5 | 46.8 | 47.8 |
| SADE | ∗ | 67.2 | 64.5 | 61.2 | 58.6 | 53.9 | 50.9 | 51.0 | 51.7 | **52.8** |
| LSC | ∗ | **68.4** | **65.8** | 62.2 | 59.2 | 53.4 | 50.4 | **51.4** | **51.8** | 52.3 |

in the neural estimator generally results in performance levels that are on par with each other. In our experiments, we default to a configuration with two layers due to its consistently strong performance.

## C.5 COMPARISON OF DIFFERENT LOSS FUNCTIONS

In this experiment, we compare the effects of using different loss functions for the training of *neural estimator*. We optimize KL-divergence by default in our implementation. We compare it with optimizing the mean squared error in Table 13. It can be seen that these two loss functions yield similar results when the test label distribution is forward and uniform. However, the KL divergence shows superior performance when the test label distribution is backward.

## C.6 VISUALIZING PREDICTED LOGITS

Figure 3 provides an intuitive visualization of the effect of *logit clipping* on predicted logits from the pre-trained model. In this figure, the left plot represents the original logit matrix, while the middle shows the results after applying logit clipping with a fixed value of $k$. The right plot displays the results of using our proposed method, which adaptively selects the optimal value of $k$ for varying test label distributions. The results show that LSC can effectively address the issue of head class suppression of pre-trained models.

Table 11: The performance improvement by our proposed test-time strategy on ImageNet-LT with various test class distributions.

| Test Dist. | Ours w/o test-time strategy | | | | Ours w/ test-time strategy | | | |
| --- | --- | --- | --- | --- | --- | --- | --- | --- |
| | Many | Med. | Few | All | Many | Med. | Few | All |
| Forward-50 | 73.6 | 60.7 | 42.6 | 70.9 | 76.0 | 58.0 | 31.6 | 73.0 (+2.1) |
| Forward-5 | 73.0 | 58.0 | 40.2 | 65.6 | 74.3 | 56.7 | 35.1 | 65.6 (+0.0) |
| Uniform | 72.0 | 57.2 | 39.9 | 60.5 | 72.0 | 57.2 | 39.9 | 60.5 (+0.0) |
| Backward-5 | 70.7 | 56.2 | 39.3 | 55.1 | 59.3 | 59.0 | 55.7 | 58.3 (+3.2) |
| Backward-50 | 69.8 | 54.4 | 38.4 | 48.4 | 57.7 | 58.7 | 55.6 | 57.3 (+8.9) |

Table 12: Effect of different layers of *neural estimator* on ImageNet-LT.

| # layers | Forward | | Uni. | Backward | |
| --- | --- | --- | --- | --- | --- |
| | 50 | 5 | 1 | 5 | 50 |
| 1 | 71.6 | 65.6 | 60.9 | 57.8 | 54.2 |
| 2 | 72.3 | 65.6 | 60.5 | 58.2 | 57.3 |
| 3 | 72.3 | 65.7 | 60.5 | 58.2 | 56.2 |
| 4 | 72.3 | 65.7 | 60.5 | 58.1 | 55.6 |
| 5 | 72.3 | 65.7 | 60.5 | 58.0 | 55.2 |

Table 13: Effect of different loss function of *neural estimator* on ImageNet-LT.

| Loss function | Forward | | Uni. | Backward | |
| --- | --- | --- | --- | --- | --- |
| | 50 | 5 | 1 | 5 | 50 |
| KL-divergence | 72.3 | 65.6 | 60.5 | 58.2 | 57.3 |
| Mean squared error | 72.3 | 65.6 | 60.5 | 55.1 | 55.8 |

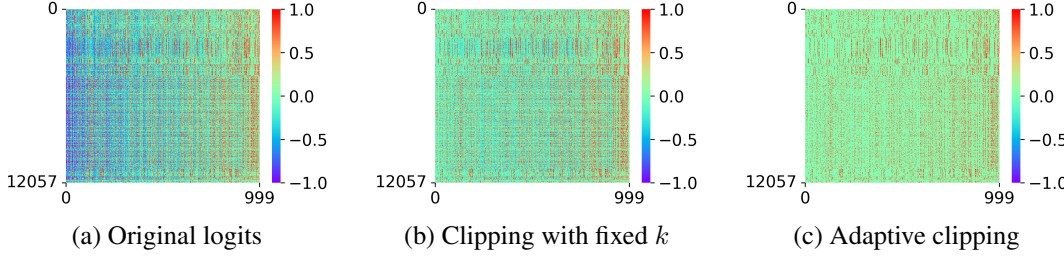

(a) Original logits      (b) Clipping with fixed $k$      (c) Adaptive clipping

Figure 3: The predicted logits before and after clipping. The experiment is conducted under the *forward-50* setting of the test data on ImageNet-LT.

## C.7 IMPACT OF HYPERPARAMETER $\lambda$ IN EQUATION (5)

We examine the impact of $\lambda$ on the final performance by selecting values from the set $\{1, 1.5, 2, 3\}$. The experiments are conducted on the ImageNet-LT dataset, and the results are presented in Table 14. Overall, we observe comparable performance across different values of $\lambda$ in the majority of cases.

## C.8 RESULTS ON "UNSEEN" TEST CLASS DISTRIBUTIONS

We are wondering if the proposed approach can tackle the test label distribution of higher class imbalance ratios since the *neural estimator* only experienced imbalance ratios in range $[1/100, 100]$. To answer this question, we test our method on the CIFAR100-LT dataset by varying the test label distribution from *Forward 100* to *Forward 200* and *Backward 100* to *Backward 200*. We compare the performance of our method LSC with the base model in Table 15. In general, the results show that

Table 14: Ablation study for $\lambda$ on ImageNet-LT.

|  | Forward | | Uni. | Backward | |
|---|---|---|---|---|---|
| $\lambda$ | 50 | 5 | 1 | 5 | 50 |
| 1 | 72.3 | 65.6 | 59.2 | 58.2 | 57.3 |
| 1.5 | 72.3 | 65.6 | 60.5 | 58.2 | 57.3 |
| 2 | 72.3 | 65.6 | 60.5 | 58.2 | 57.3 |
| 3 | 70.9 | 65.6 | 60.5 | 55.0 | 57.3 |

our method consistently improves the base model across all settings. Notably, the performance of LSC remains stable even as the test label distribution becomes more imbalanced.

Table 15: Results for higher imbalance ratios on CIFAR100-LT.

| Imbalance ratio | 100 | 120 | 140 | 160 | 180 | 200 |
|---|---|---|---|---|---|---|
| **w/o** LSC | | | | | | |
| *Forward* | 65.0 | 65.5 | 65.9 | 66.3 | 66.2 | 66.2 |
| *Backward* | 37.8 | 37.3 | 37.1 | 36.7 | 36.4 | 36.3 |
| **w/** LSC | | | | | | |
| *Forward* | 70.5 | 71.3 | 71.8 | 72.2 | 72.3 | 72.6 |
| *Backward* | 48.0 | 47.9 | 48.1 | 48.0 | 47.9 | 48.0 |

# D  ON THE CLASS-PRIOR BASED RE-SAMPLING

## D.1  EFFICIENCY OF RE-SAMPLING

We demonstrate that the re-resampling module used to construct "training data" for our proposed *nueral estimator* can be implemented efficiently through the following Python code. Specifically, given each class prior, we first determine the number of samples for each class for simulation. Subsequently, we sample a subset of training samples for each class by pre-dividing the samples in the entire training set into their respective class pools. In general, the time complexity of the re-sampling process is approximately $O(QN)$, where $Q$ is the number of times of re-sampling and $N$ is the number of training samples. By default, we set $Q = 2 \times 10^3$ in all experiments and the re-sampling process can be done in seconds for training data of size $N = 10^6$ on modern computers. We also monitor the time consumption of the entire process of GBBSE, it takes 12.78 seconds and 355.15 seconds on CIFAR100-LT and ImageNet-LT, respectively.

```python
def multiple_subset_resampling(imbalance_ratio_list, img_idx_each_cls):
  """sample multiple subsets from a list of class pirors"""
  subset_list = []
  for imbalance_ratio in imbalance_ratio_list:
    selected_idx_list = subset_resampling(imbalance_ratio,
                                           img_idx_each_cls)
    subset_list.append(selected_idx_list)
  return subset_list

def subset_resampling(imbalance_ratio, img_idx_each_cls):
  """sample a subset from a given class prior"""
  img_num_per_cls = produce_num_per_cls(imbalance_ratio)
  selected_idx_list = []
  for the_class, the_img_num in enumerate(img_num_per_cls):
    idx = img_idx_each_cls[the_class]
    selected_idx = torch.multinomial(torch.ones(idx.size(0)), the_img_num
                                     , replacement=True)
      selected_idx_list.append(selected_idx)
  return selected_idx_list
```

## D.2    HOW DOES THE NUMBER OF SAMPLING SUBSETS IMPACT THE PERFORMANCE?

We explore the impact of the number of sampled subsets on the ImageNet-LT dataset. We conduct simulations with subsets ranging from $\{400, 1000, 2000, 4000\}$ and report the test accuracy in Table 16. Generally, we observe an improvement in test accuracy with an increase in the number of simulated subsets. This is attributed to the aim of ensuring that the training set for $g_\theta$ covers diverse label distributions. However, we only sample 2000 subsets for a balance between effectiveness and efficiency.

Table 16: Impact of the number of sampled subsets on ImageNet-LT.

| # layers | Forward | | Uni. | Backward | |
|---|---|---|---|---|---|
| | 50 | 5 | 1 | 5 | 50 |
| 4000 | 72.3 | 65.6 | 60.5 | 58.0 | 57.7 |
| 2000 | 72.3 | 65.6 | 60.5 | 58.3 | 57.3 |
| 1000 | 72.2 | 65.6 | 60.5 | 58.2 | 56.8 |
| 400 | 70.9 | 65.6 | 61.0 | 57.1 | 52.7 |

