# OpenReview forum: "Learning Label Shift Correction for Test-Agnostic Long-Tailed Recognition"
_ICLR.cc/2024/Conference — Submitted to ICLR 2024_

### Official Review · Reviewer_zgiF · 2023-10-30

**Soundness:** 3 good
**Presentation:** 3 good
**Contribution:** 3 good
**Rating:** 6
**Confidence:** 4

**Summary:**

This paper considers test-agnostic long-tailed recognition, where the training distribution is long-tailed and the test distribution can be different from the training distribution, not necessarily uniform but unknown. Based on the theoretical results showing that if one can estimate the test label distribution, it can be possibly used to reduce generalization error even with the distribution shift during training and test time, this paper proposes a label shift correction (LSC) method. LSC estimates the test label distribution using the trained neural estimator and the predicted logits for test data using the pre-trained model. The neural estimator is trained to minimize the distance between the estimated distribution using the logit output of the pre-trained model and the true label distribution, for various label distributions generated by sampling training data while varying class priors. Also, the adaptive logit clipping is introduced to clipping spurious model outputs. The authors demonstrate the efficacy of the proposed method on CIFAR-10/100-LT and ImageNet-LT and also show that the proposed method can be combined with existing LT method to further boost the performance.

**Strengths:**

- Demonstrated that estimating the test label distribution and correcting prediction of the model using the estimated test label distribution is an effective way of boosting the classification performance in long-tail learning.
- Proposed a simple yet effective method to estimate the test label distribution by training a neural estimator using various label distributions generated by sampling training data while varying class priors. Proposed an adaptive logit clipping method that turns out to be important in adjusting the predictions and effectively choosing the promising $k$ in an adaptive fashion.

**Weaknesses:**

- The proposed method was not evaluated on general test label distributions, but rather on three limited types, forward, uniform and backward. How does the performance of the proposed method change as encountering more general label distribution shift?
- The effectiveness of neural estimator in correcting the label distribution shift may highly depend on how many ($Q$ in Alg. 1) and diverse class priors have been encountered during the neural estimator. The authors need to explain more about this and the corresponding overhead in training. For example, when the test distribution has imbalance ratio (backward with 200), can the proposed method, which only experienced the imbalance ratio of range [1/100,100], correctly estimate the test label distribution?

**Questions:**

- More elaborations on Eq. (5) is needed. First, it is unclear how $\hat{Z}^h_{ij}$ and $\hat{Z}^t_{ij}$ are defined. The authors comment that $\hat{Z}^h$ and $\hat{Z}^t$ are the sum of head and tail parts in $\hat{Z}$. Then, what does index $j$ mean in $\hat{Z}^h$ and $\hat{Z}^t$? Can the authors elaborate why the optimal $k$ maximizing (5) results in a higher value for the backward case and lower for the forward case?
- The proposed method was not evaluated on general test label distributions, but rather on three limited types, forward, uniform and backward. How does the performance of the proposed method change as encountering more general label distribution shift?
- The effectiveness of neural estimator in correcting the label distribution shift may highly depend on how many ($Q$ in Alg. 1) and diverse class priors have been encountered during the neural estimator. The authors need to explain more about this and the corresponding overhead in training. For example, when the test distribution has imbalance ratio (backward with 200), can the proposed method, which only experienced the imbalance ratio of range [1/100,100], correctly estimate the test label distribution?

---

> ### Author Response · Authors · 2023-11-15
> **Official Comment by Authors (Part 1)**
>
> We are grateful for the valuable reviews, and for the positive comments such as **the method is simple yet effective**. We will address the reviewer's concerns below.
>
> **Concern #1: How does the performance change as encountering more general label distribution shift?**
>
> **Response:**
>   In this paper, we delve into the test-agnostic long-tailed recognition problem, aligning with previous research efforts such as SADE [NeurIPS 2021], LADE [CVPR 2021], and BalPoe [CVPR 2023]. To the best of our knowledge, no existing long-tail learning method can generalize to arbitrary test label distributions—a limitation inherent in the no-free-lunch theorem. To address this challenge, a pivotal issue is how to effectively simulate diverse label distributions, a facet that has not been extensively explored in the field. By developing the capability to simulate any label distribution, we believe our proposed method can encompass various simulations into the training data, thereby learning to estimate arbitrary label distributions.
>
> **Concern #2: The effectiveness of the training of neural estimator.**
>
>   **Response:**
>   We show that the training of our proposed neural estimator can be done efficiently. We first elaborate on the simulation of training data for the neural estimator. Given a class prior, we determine the corresponding number of samples for each class for simulation. Subsequently, we swiftly sample a subset of training samples for every class by pre-dividing the samples in the entire training set into their respective class pools. In general, the time complexity of the re-sampling process is approximately $O(QN)$, where *$Q$* is the number of times of re-sampling and $N$ is the number of training samples. By default, we set $Q=2\times10^3$ in all experiments and the re-sampling process can be done in seconds for training data of size $N=10^6$ on modern computers. We also monitor the time consumption of the entire training process of the neural estimator, it takes *12.78* seconds and *355.15* seconds on CIFAR100-LT and ImageNet-LT, respectively. To demonstrate the effectiveness of the simulation of different label distributions, we attach the `Python` code for reference.
>
>   ```python
>   def multiple_subset_resampling(imbalance_ratio_list, img_idx_each_cls):
>     """sample multiple subsets from a list of class priors"""
>     subset_list = []
>     for imbalance_ratio in imbalance_ratio_list:
>       selected_idx_list = subset_resampling(imbalance_ratio, img_idx_each_cls)
>       subset_list.append(selected_idx_list)
>     return subset_list
>
>   def subset_resampling(imbalance_ratio, img_idx_each_cls):
>     """sample a subset from a given class prior"""
>     img_num_per_cls = produce_num_per_cls(imbalance_ratio)
>     selected_idx_list = []
>     for the_class, the_img_num in enumerate(img_num_per_cls):
>       idx = img_idx_each_cls[the_class]
>       selected_idx = torch.multinomial(torch.ones(idx.size(0)), the_img_num, replacement=True)
>     	selected_idx_list.append(selected_idx)
>     return selected_idx_list
>   ```
>
> **Concern #3: In Eq. (5), what does index $j$ mean?**
>
> **Response:**
>   We are sorry for the confusion and correct Eq. (5) as follows:
>
>   $$  k = \arg \max _ {k \in \mathcal{K}} \mathbb{I}(\pi _ 0^{h} > \lambda \pi _ 0^{t}) \widehat{Z}^{h} + \mathbb{I}(\pi _ 0^{h} < \lambda \pi _ 0^{t}) \widehat{Z}^{t} \quad s.t. \quad \widehat{Z}=logitClip({Z}, k)$$
>
> **Concern #4: Why the optimal $k$ maximizing Eq. (5) results in a higher value for the backward case and lower for the forward case?**
>
> **Response:**
> The rationale behind Eq. (5) is twofold.
>
> Firstly, we observe that most long-tail learning methods generate logits that exhibit a bias towards tail classes. Specifically, the model may produce large (positive) logits for tail classes even when the ground truth corresponds to a head class. Conversely, the produced logits for head classes can be small (negative) values, excluding the ground-truth class. Consequently, without a higher value of $k$, it becomes easy to observe the backward label distribution, as tail classes dominate head classes in the logits matrix. However, we require a lower value of $k$ to clip negative logits of head classes, enabling the identification of a forward label distribution. This observation is visually depicted in Figure 2(b) of the paper.
>
> Secondly, we utilize Eq. (5) to determine the value of $k$ because the predicted probabilities (i.e., $\pi _ 0$) can provide a rough sketch of the true label distribution. This observation is illustrated in Figure 2(a) of the paper. Combining these observations, Eq. (5) is capable of automatically searching for a suitable value of $k$.

---

> ### Author Response · Authors · 2023-11-15
> **Official Comment by Authors (Part 2)**
>
> **Concern #5: Can the proposed method, which only experienced the imbalance ratio of range [1/100,100], correctly estimate the test label distribution?**
>
> **Response:**
> This is a very intriguing question. To answer this question, we test our method on the CIFAR100-LT dataset by varying the test label distribution from Forward100 to Forward200 and Backward100 to Backward200. We compare the performance of our method LSC with the base model. In general, the results show that our method consistently improves the base model across all settings. Notably, the performance of LSC remains stable even as the test label distribution becomes more imbalanced.
>
>   | **w/ LSC**  | 100 | 120 | 140 | 160 | 180 | 200 |
>   | ----------- | ---- | ---- | ---- | ---- | ---- | ---- |
>   | Forward   | 70.5 | 71.3 | 71.8 | 72.2 | 72.3 | 72.6 |
>   | Backward  | 48.0 | 47.9 | 48.1 | 48.0 | 47.9 | 48.0 |
>   | **w/o LSC** |   |   |   |   |   |   |
>   | Forward   | 65.0 | 65.5 | 65.9 | 66.3 | 66.2 | 66.2 |
>   | Backward  | 37.8 | 37.3 | 37.1 | 36.7 | 36.4 | 36.3 |

---

> > ### Comment · Reviewer_zgiF · 2023-11-22
> >
> > Thank the authors for providing detailed answers. Most of my concerns are resolved, though my first question about dealing with more general label distribution shift still remains open. I will have further discussion with other reviewers and AC. Thank you.

---

### Official Review · Reviewer_pK6C · 2023-10-31

**Soundness:** 2 fair
**Presentation:** 2 fair
**Contribution:** 2 fair
**Rating:** 5
**Confidence:** 3

**Summary:**

This paper proposes a simple yet efficient method called label shift correction (LSC) to estimate a more accurate test label distribution for addressing the problem of test-agnostic long-trailed learning. The proposed method is motivated by the theoretical insight, which indicates that precise estimation of the true test label distribution can reduce the generalization error. Compared with BBSE, they propose a generalized version called GBBSE to align the predicted test label distribution with the true test label distribution via a family of parameterized label distribution estimation functions.

**Strengths:**

1. **[The question is important in practice.]** In the real world, the condition of the test label distribution is often infeasible to obtain. Thus, it is important to propose a general method for handling different distribution-shift scenarios.
2. **[The motivation of this paper is clear.]** The authors have theoretically certified that better alignment between the predicted test label distribution and the true test label distribution leads to lower generalization error. Therefore, readers can clearly understand the reasons why they focus on label shift correction.

**Weaknesses:**

1. **[Writting about the proposed method is confusing.]** In my view, there are several points in Section 3 making me confused. (1) Your proposed method is LSC, but I notice that you use a lot of space to introduce GBBSE. So what is the connection between LSC and GBBSE? (2) What does $g_{\theta}$ refer to in your experiments? Is $g_{\theta}$ equal to NeuralEstimator? (3) Since the training dataset is imbalanced, how can you sample a subset of the training dataset when its smallest class should be the biggest class in the subset?

2. **[The highlight of the proposed method is not clear.]** The goal of LSC is to find an estimator $g$ that aligns the predicted test label distribution with the true. But in Line 7 of the pseudo-code, $g_{\theta}$ aims to align subset label distribution with the prior global label distribution on the training dataset. So can you explain which step achieves this goal?

3. **[Your theoretical analysis cannot explain why your proposed method can align the discrepancy between the predicted and the true test label distributions.]** You have mentioned two theorems in your paper. However, both of them only certify that a smaller gap between predicted and the true test label distributions leads to a lower generalization error. They do not explain why your proposed method can effectively estimate the true test label distribution.

**Questions:**

1. Please answer the questions mentioned in the first part of Weakness.
2. Can you explain which step achieves the alignment?
3. Can you theoretically analyze why this method can align the predicted and the true test label distribution?
4. What is the effect of $\lambda$ on the final performance? Why do you choose the value of $\lambda$ as 1.5?
5. Does the number of sampling subsets of the training dataset affect the final performance?

---

> ### Author Response · Authors · 2023-11-15
> **Official Comment by Authors (Part 1)**
>
> We appreciate the valuable reviews and the positive comments highlighting the importance of the question and the clarity of the motivation. Below, we will address the concerns raised by the reviewer.
>
> **Concern #1: The connection between LSC and GBBSE?**
>
> **Response:** Sorry for any confusion. This paper presents a general label-shift estimation framework called GBBSE. We then propose a practical implementation under the framework of GBBSE and the implemented method is called LSC. We will improve the writing in the next version soon.
>
>
> **Concern #2: Is $g _ \theta$ equal to *NerualEstimator*?**
>
> **Response:** Yes, we refer to $g _ \theta$ as the *NerualEstimator* in our paper.
>
>
> **Concern #3: Since the training dataset is imbalanced, how can you sample a subset of the training dataset when its smallest class should be the biggest class in the subset?**
>
> **Response:**
> Given a class prior, we utilize either undersampling or oversampling for simulation. In cases where additional training data is required for the smallest class, we implement resampling of the training data with replacement. To enhance diversity, we incorporate data augmentation strategies, such as RandAugment or AutoAugment, which generate distinct copies of the same image. To elucidate the sampling process, we provide the attached `Python` code for reference:
>
>   ```python
>   def multiple_subset_resampling(imbalance_ratio_list, img_idx_each_cls):
>     """sample multiple subsets from a list of class priors"""
>     subset_list = []
>     for imbalance_ratio in imbalance_ratio_list:
>       selected_idx_list = subset_resampling(imbalance_ratio, img_idx_each_cls)
>       subset_list.append(selected_idx_list)
>     return subset_list
>
>   def subset_resampling(imbalance_ratio, img_idx_each_cls):
>     """sample a subset from a given class prior"""
>     img_num_per_cls = produce_num_per_cls(imbalance_ratio)
>     selected_idx_list = []
>     for the_class, the_img_num in enumerate(img_num_per_cls):
>       idx = img_idx_each_cls[the_class]
>       selected_idx = torch.multinomial(torch.ones(idx.size(0)), the_img_num, replacement=True)
>     	selected_idx_list.append(selected_idx)
>     return selected_idx_list
>   ```
>
>
>
> **Concern #4: Can you explain which step achieves the alignment?**
>
> **Response:** As illustrated in the pseudo-code, the training of $g _ \theta$ is conducted on simulated subsets with various label distributions, rather than the global label distribution. The objective is to minimize the discrepancy between the output of $g _ \theta$ and the ground-truth label distribution of our simulated subsets. Ultimately, the well-trained $g _ \theta$ is able to generalize to the test data and estimate the test label distribution. We will modify the pseudo-code to make the notations more clear.
>
>
> **Concern #5: Can you theoretically analyze why this method can align the predicted and the true test label distribution?**
>
> **Response:**
>   Theorem 3.2 in our paper ensures that we can acquire a sufficiently trained base model, i.e., $\epsilon _ {L}(g^{\star})$ is small. Assuming that the error gap $\epsilon _ {L}(g _ {\theta}) - \epsilon _ {L}(g^{\star})$ can be bounded through the training of $g _ {\theta}$ using generated multiple subsets (ensured by the Bayes-risk consistency of the training), we can approximate the test label distribution precisely to adjust the model, especially when the test sample size is sufficiently large.
>   More precisely, assuming that the multiple subsets are obtained by *i.i.d* sampling, we can bound the estimation error through classical generalization theories.
>
>   However, in the context of long-tail learning, generating multiple subsets for training $g _ {\theta}$ involves oversampling for tail classes, disrupting the *i.i.d* properties of sampling. This departure from *i.i.d* conditions complicates the application of traditional machine learning generalization analysis. To the best of our knowledge, analyzing the generalization bound in this scenario without additional assumptions proves to be extremely challenging.
>
>   **Concern #6: What is the effect of $\lambda$ on the final performance?**
>
> **Response:**
>   We examine the impact of $\lambda$ on the final performance by selecting values from the set $\\{1, 1.5, 2, 3\\}$. The experiments are conducted on the ImageNet-LT dataset, and the results are presented in the table. Overall, we observe comparable performance across different values of $\lambda$ in the majority of cases.
>
>   |$\lambda$ | Forward50 | Forward5  | Uniform | Backward5  | Backward50 |
>   |:---------:|:----:|:----:|:----:|:----:|:----:|
>   | 1     | 72.3 | 65.6 | 59.2 | 58.2 | 57.3 |
>   | 1.5    | 72.3 | 65.6 | 60.5 | 58.2 | 57.3 |
>   | 2     | 72.3 | 65.6 | 60.5 | 58.2 | 57.3 |
>   | 3     | 70.9 | 65.6 | 60.5 | 55.0 | 57.3 |

---

> ### Author Response · Authors · 2023-11-15
> **Official Comment by Authors (Part 2)**
>
> **Concern #6: Does the number of sampling subsets of the training dataset affect the final performance?**
>
> **Response:** Thank you for this interesting question. We explore the impact of the number of sampled subsets on the ImageNet-LT dataset. We conduct simulations with subsets ranging from $\\{400, 1000, 2000, 4000\\}$ and analyze the resulting final performance. Generally, we observe an improvement in test accuracy with an increase in the number of simulated subsets. This is attributed to the aim of ensuring that the training set for $g _ \theta$ covers diverse label distributions. However, we only sample 2000 subsets for a balance between effectiveness and efficiency.
>
>   | # of subsets | Forward50 | Forward5  | Uniform | Backward5  | Backward50 |
>   |:-----------:|:----:|:----:|:----:|:----:|:----:|
>   | 4000    | 72.3 | 65.6 | 60.5 | 58.0 | 57.7 |
>   | 2000    | 72.3 | 65.6 | 60.5 | 58.3 | 57.3 |
>   | 1000    | 72.2 | 65.6 | 60.5 | 58.2 | 56.8 |
>   | 400     | 70.9 | 65.6 | 61.0 | 57.1 | 52.7 |

---

> > ### Comment · Reviewer_pK6C · 2023-11-21
> >
> > Dear authors,
> >
> > Thank you for your reply. I appreciate the effort from the authors in answering our questions with further explanations and additional empirical verifications. I will make my final decision after communicating with AC and the other reviewers during the AC-Reviewer discussion period.
> >
> > Best regards,
> >
> > Reviewer pK6C

---

### Official Review · Reviewer_nfhw · 2023-10-31

**Soundness:** 2 fair
**Presentation:** 3 good
**Contribution:** 2 fair
**Rating:** 6
**Confidence:** 3

**Summary:**

Motivated by the observation that real-world test distributions are not uniform, the paper proposes label shift correction (LSC), a method for test-agnostic long-tail learning. LSC does not require access to the truth test label distribution, but instead estimates it through a generalization of Black Box Shift Estimation (BBSE) with logit clipping. LSC empirically achieves SOTA accuracy on long-tail image classification tasks under different test distributions, and works well with existing long-tail learning methods.

**Strengths:**

The method is motivated from a Bayesian perspective. LSC exhibits strong empirical performance over baselines assuming a uniform target label distribution. Furthermore, it synergizes with existing long-tail learning methods. Moreover, as shown in Table 4, it consistently improves the performance for any test distribution. Last but not least, the paper provides ample ablation study to dissect LSC.

**Weaknesses:**

BBSE is not the best performing in label-shift estimation. In particular, [1] points out that an MLE-based approach called MLLS dominates BBSE. The statistical inefficiency is confirmed by LSC's suboptimal performance on "Backward" test distributions (it should be as good as "Forward"), so the authors' choice of BBSE seems unjustified. Moreover, the GBBSE requires re-sampling multiple subsets, which might take non-trivial time to train.

[1]: A Unified View of Label Shift Estimation (https://arxiv.org/abs/2003.07554)

**Questions:**

You employ logit clipping to address overconfident logits for tail classes. Is that related to the miscalibration of neural networks? While BBSE itself does not require well-calibrated logits, miscalibration might be an issue when you clip unnormalized logits to zero. I suggest looking into bias-corrected temperature scaling (BCTS), which is a component of the MLLS method discussed in [1]. My intuition is that you don't need logit clipping after calibration.

[2]: On Calibration of Modern Neural Networks (https://arxiv.org/abs/1706.04599)
[3]: Maximum Likelihood with Bias-Corrected Calibration is Hard-To-Beat at Label Shift Adaptation (http://proceedings.mlr.press/v119/alexandari20a/alexandari20a.pdf)

---

> ### Author Response · Authors · 2023-11-15
> **Official Comment by Authors (Part 1)**
>
> We express gratitude to the reviewer for their insightful reviews and encouraging remarks, including **strong empirical performance** and **ample ablation study**. Below, we address the concerns raised by the reviewer.
>
> **Concern #1: suboptimal performance on "Backward" test distributions compared to "Forward".**
>
> **Response:**
> Indeed, nearly every long-tail learning method experiences a decline in performance when transitioning from a "Forward" to a "Backward" test distribution. In our setup, where the training data adheres to a "Forward" long-tail distribution, the model demonstrates robust generalization when the test distribution aligns with the "Forward" pattern. Conversely, the model's performance may falter when faced with a "Backward" test distribution, as it lacks exposure to ample samples from tail classes during training. While some methods can be employed to address the challenges of "Backward" performance, their effectiveness remains suboptimal. In response to this issue, we introduce LSC in this paper, specifically designed to address label distribution shifts, and it surpasses previous methods by a significant margin. Acknowledging the no-free-lunch theorem, we recognize the impossibility of achieving perfect compensation for data scarcity and generalizing seamlessly to "Backward" as well as "Forward".
>
> From a theoretical standpoint, the implications of Theorem 3.1 in our paper align with this issue. For the sake of simplicity, we assume the true posterior distribution $P(Y \mid X)$ is one-hot, and we focus on the first term in Theorem 3.1 as follows:
> $$
> \begin{aligned}
> \left \| \widehat{P}(Y\mid X) - P _ {\mathcal{D} _ {S}}(Y\mid X) \right \| _ {L^{1}, w} &= \mathbb{E} _ {x \sim \mathcal{D} _ {S}} \left[ \sum _ {y \in \left[K\right]} w _ {y} \left |\widehat{P}(y\mid x) - P _ {\mathcal{D} _ {S}}(y\mid x) \right | \right] = \sum _ {i\in [K]} P _ {S}(Y=i) \mathbb{E} _ {x \sim \mathcal{D} _ {S}(\cdot \mid Y=i)} \left[ \sum _ {y \in \left[K\right]} w _ {y} \left |\widehat{P}(y\mid x) - P _ {\mathcal{D} _ {S}}(y\mid x) \right | \right]
>  \end{aligned}
> $$
> Assume the label shift estimation is perfect, i.e., $\widehat{P} _ {T}(Y)=P _ {T}(Y)$, then we have:
> $$
> \begin{aligned}
> P _ {S}(Y=k) \mathbb{E} _ {x \sim \mathcal{D} _ {S}(\cdot \mid Y=k)} \left[ \sum _ {y \in \left[K\right]} w _ {y} \left |\widehat{P}(y\mid x) - P _ {\mathcal{D} _ {S}}(y\mid x) \right | \right] \ge P _ {T}(Y=k) \mathbb{E} _ {x \sim \mathcal{D} _ {S}(\cdot \mid Y=k)} \left[\left(1 - \widehat{P}(Y=k\mid x)\right) \right]
> \end{aligned}
> $$
> During the training of the base model, we minimize the empirical cross-entropy loss on the training set to address the aforementioned term. However, there exists an inherent gap between empirical error and generalization error, which is linked to the number of samples. In prior studies, including [1, 2], this gap is typically bounded by $O(\frac{1}{\sqrt{N}})$, where we have:
> $$
> \begin{aligned}
> \mathbb{E} _ {x \sim \mathcal{D} _ {S}(\cdot \mid Y=k)} \left[\left(1 - \widehat{P}(Y=k\mid x)\right) \right] & \le \mathbb{E} _ {x \sim \mathcal{D} _ {S}(\cdot \mid Y=k)} \left[-\log \widehat{P} (Y=k\mid x) \right] \le \sum _ {x \in D _ {S}^{k}}\frac{-\log \widehat{P} (Y=k\mid x)}{N _ {k}} + O(\frac{1}{\sqrt{N _ {k}}})
> \end{aligned}
> $$
> Hence, $\mathbb{E} _ {x \sim \mathcal{D} _ {S}(\cdot \mid Y=k)} \left[\left(1 - \widehat{P}(Y=k\mid x)\right) \right]$ increases as the number of $k$-th class training samples decreases. Under such observation, we have:
> $$
> \begin{aligned}
> \left \| \widehat{P}(Y\mid X) - P _ {\mathcal{D} _ {S}}(Y\mid X) \right \| _ {L^{1}, w} & \ge \sum _ {k \in [K]} P _ {T}(Y=k)\mathbb{E} _ {x \sim \mathcal{D} _ {S}(\cdot \mid Y=k)} \left[\left(1 - \widehat{P}(Y=k\mid x)\right) \right]
> \end{aligned}
> $$
>  Given that we have $N _ {1} \geq N _ {2}\geq \dots \geq N _ {K}$, it is evident that the error gradually increases as the test data distribution leans toward the tail class.
> This phenomenon is solely linked to the long-tailed distribution of the training set, assuming perfect test label distribution prediction has been achieved during the derivation process. For a more in-depth exploration of this phenomenon, refer to [1, 2]. Numerous long-tail learning methods, such as [3, 4, 5], have been introduced to address this issue. Notably, LSC can seamlessly complement existing long-tail learning methods.
>
> [1]. A Unified Generalization Analysis of Re-Weighting and Logit-Adjustment for Imbalanced Learning.
> https://arxiv.org/pdf/2310.04752.pdf
>
> [2]. Sharp error bounds for imbalanced classification: how many examples in the minority class?
> https://arxiv.org/abs/2310.14826.pdf
>
> [3]. Long-tailed recognition by routing diverse distribution-aware experts.
> https://arxiv.org/abs/2010.01809.pdf
>
> [4]. Parametric contrastive learning.
> https://arxiv.org/abs/2107.12028.pdf
>
> [5]. Nested collaborative learning for long-tailed visual recognition.
> https://arxiv.org/abs/2203.15359.pdf

---

> ### Author Response · Authors · 2023-11-15
> **Official Comment by Authors (Part 2)**
>
> **Concern #2: the choice of BBSE seems unjustified.**
>
> **Response:**
> In this paper, we present GBBSE -- a more general test distribution estimation framework built upon BBSE. We improve the theoretical results of BBSE in the context of long-tail scenarios. Furthermore, in our implementation, we enhance BBSE by substituting the confusion matrix estimation with the training of a neural estimator. While acknowledging that BBSE may not be the optimal choice for label-shift estimation, we opt for it due to its simplicity and robust theoretical properties, such as an appealing convergence rate of the size of the validation set $O(\frac{\log N}{N})$. It is worth noting that alternative approaches like MLLS also necessitate an additional validation set, a resource typically unavailable in long-tail learning scenarios. We plan to discuss this with MLLS and BCTS in the next version.
>
> **Concern #3: GBBSE requires re-sampling multiple subsets, which might take non-trivial time to train.**
>
> **Response:**
> We demonstrate that the re-resampling module can be implemented efficiently through the attached `Python` code. Specifically, given each class prior, we first determine the number of samples for each class for simulation. Subsequently, we sample a subset of training samples for each class by pre-dividing the samples in the entire training set into their respective class pools. In general, the time complexity of the re-sampling process is approximately $O(QN)$, where *$Q$* is the number of times of re-sampling and $N$ is the number of training samples. By default, we set $Q=2\times10^3$ in all experiments and the re-sampling process can be done in seconds for training data of size $N=10^6$ on modern computers. We also monitor the time consumption of the entire process of GBBSE, it takes *12.78* seconds and *355.15* seconds on CIFAR100-LT and ImageNet-LT, respectively.
>
>   ```python
>   def multiple_subset_resampling(imbalance_ratio_list, img_idx_each_cls):
>     """sample multiple subsets from a list of class pirors"""
>     subset_list = []
>     for imbalance_ratio in imbalance_ratio_list:
>       selected_idx_list = subset_resampling(imbalance_ratio, img_idx_each_cls)
>       subset_list.append(selected_idx_list)
>     return subset_list
>
>   def subset_resampling(imbalance_ratio, img_idx_each_cls):
>     """sample a subset from a given class prior"""
>     img_num_per_cls = produce_num_per_cls(imbalance_ratio)
>     selected_idx_list = []
>     for the_class, the_img_num in enumerate(img_num_per_cls):
>       idx = img_idx_each_cls[the_class]
>       selected_idx = torch.multinomial(torch.ones(idx.size(0)), the_img_num, replacement=True)
>     	selected_idx_list.append(selected_idx)
>     return selected_idx_list
>   ```
>
> **Concern #4: You employ logit clipping to address overconfident logits for tail classes. Is that related to the miscalibration of neural networks?**
>
> **Response:**
>   To validate the reviewer's assumption, we report both the Expected Calibration Error (ECE) and Accuracy of LSC and its counterpart without using logit clipping. We implement LSC on two distinct base models, namely, NCL and RIDE. The experiments are conducted on the CIFAR100-LT dataset. Overall, the results do not indicate a strong correlation between miscalibration and logit clipping. Specifically, NCL exhibits good calibration with a low ECE. However, NCL+LSC without logit clipping shows poor performance on both "Forward" and "Uniform" test distributions. This is attributed to the fact that the logits generated by NCL for head classes are still dominated by tail classes. In contrast, NCL+LSC with logit clipping significantly improves NCL performance in both "Forward" and "Backward" cases. Unlike NCL, RIDE is a model with poor calibration, characterized by a high ECE. Remarkably, LSC with logit clipping is still effective in enhancing its performance across all cases. These results highlight the efficacy of our proposed method for various base models, whether they exhibit good or poor calibration.
>
> | CIFAR100-LT *(ECE/ACC)*      | Foward50     | Foward5     |Uniform    | Backward5     | Backward50    |
> |:-----------------------:|:------------:|:----------:|:----------:| ----------:|:----------:|
> | NCL        | 0.04 (63.7)  | 0.06 (57.3) | 0.08 (51.9) | 0.11 (46.0) | 0.14 (39.6) |
> | NCL+LSC w/o logit clipping | 0.05 (61.6)  | 0.07 (51.2) | 0.08 (47.1) | 0.12 (46.4) | 0.08 (49.6) |
> | NCL+LSC w/ logit clipping          | 0.05 (68.1)  | 0.07 (58.4) | 0.08 (51.9) | 0.12 (46.0) | 0.08 (48.3) |
> | RIDE        | 0.81 (64.1)  | 0.78 (55.9) | 0.76 (48.6) | 0.73 (40.8) | 0.70 (31.5) |
> | RIDE+LSC w/o logit clipping | 0.85 (66.5)  | 0.81 (53.1) | 0.78 (41.9) | 0.75 (31.8) | 0.72 (19.5) |
> | RIDE+LSC w/ logit clipping          | 0.82 (66.2)  | 0.79 (56.2) | 0.76 (48.6) | 0.73 (41.3) | 0.70 (33.2) |

---

> ### Comment · Reviewer_nfhw · 2023-11-23
>
> Thanks for the clarification, analysis, and experiments. Most of my concerns have been addressed, so I have increased my rating to 6. I look forward to your future work.

---

> > ### Author Response · Authors · 2023-11-23
> > **Reply to Reviewer nfhw**
> >
> > Thanks so much for your time and positive feedback! We are encouraged and will continue to polish our work.

---

### Author Response · Authors · 2023-11-19
**We look forward to your feedback**

Dear Reviewers,

We sincerely appreciate all of you for your time and efforts. Your valuable comments are very helpful in improving our paper. In addition to the responses, **we have also updated our PDF manuscript** to accommodate your concerns. The following are the updates:

- We cite two missing related papers suggested by the reviewer, i.e., *A Unified View of Label Shift Estimation* and *Maximum Likelihood with Bias-Corrected Calibration is Hard-To-Beat at Label Shift Adaptation*
- We include the Python code for subset re-sampling as well as its efficiency study when training the proposed *neural estimator* in **Appendix D.1**
- We discuss the effect of $\lambda$ on the final performance in **Appendix C.7**
- We study how the number of sampling subsets of the training dataset affects the final performance in **Appendix D.2**
- We update the definition of *Eq. (5)* to avoid any confusion and rectify incorrect notations
- We analyze if the proposed method, which only experienced the imbalance ratio of range [1/100,100], can correctly estimate the test label distribution of higher imbalance ratios in **Appendix C.8**

Moreover, it is important for us to know whether our responses have addressed your concerns, and we look forward to receiving your further comments.

Best Regards,

Authors

---

### Meta-Review · Area_Chair_9WeQ · 2023-12-07

**Metareview:**

This paper studies learning under a long-tail label distribution. The proposed method is based on data re-sampling and data augmentation, motivated by the difficulty of estimating the confusion matrix in the classic BBSE label shift estimation problem. The idea is straightforward and the experiments show the method's effectiveness in typical long-tail settings.

Strengths: all reviewers agree that the problem is important, the motivation is very clear, and the proposed method is simple but effective.

Weaknesses: the major concern that remains is the relation to the general label shift literature. Given the close connection to label shift and the key motivation that comes from BBSE, it would be nice to have comparison with more methods in that domain under general label shift.

**Justification For Why Not Higher Score:**

With important remaining concerns, I feel this paper can be further improved to cover more grounds by considering the general label shift scenarios.

**Justification For Why Not Lower Score:**

N/A

---

### Decision · Program_Chairs · 2024-01-16

Reject